# Pick and Adapt: An Iterative Approach for Source-Free Domain Adaptation

## Abstract

Domain adaptation plays a pivotal role in deploying models when the inference data distribution is different from the training data. It becomes particularly challenging in source-free domain adaptation (SFDA) scenario, where access to the source domain data is restricted due to data privacy concerns. To tackle such cases, existing approaches often resort to generating source-like data for standard unsupervised domain adaptation or endeavor to fine-tune a model pre-trained on a source domain using self-supervised training techniques. Instead, our approach strikes a different path by theoretically analyzing an empirical risk bound for SFDA. We identify the population risk and domain drift as the major factors from the risk bound. Subsequently, we introduce a top-k importance sampling to purify the pseudo labels and thus reduce the population risk. We further present a nearest neighbor voting-based semantic domain alignment to mitigate the domain drift. An iterative optimization is finally proposed to combine the above two steps for multiple rounds. Extensive experiments across three widely applied domain adaptation datasets, i.e., Office-Home, DomainNet, and VisDA-C, demonstrate the consistently advantageous performance over the state-of-the-art methods.

## 1 Introduction

Domain adaptation (DA) has shown wide applications in machine learning and computer vision tasks, such as image or video recognition (Liu et al., 2021; Sahoo et al., 2021), segmentation (Chen et al., 2022b), and object detection (Li et al., 2022). Many popular methods have presented large success in above mentioned applications, e.g., adversarial learning (Goodfellow et al., 2020), self-supervised learning (Jing & Tian, 2020), and self-training (Zou et al., 2019). However, data privacy protection has emerged as a critical concern, with legislation such as the General Data Protection Regulation (GDPR) in Europe aimed at safeguarding individuals' information. This new focus on privacy presents a unique challenge for DA where source domain data is no longer accessible. Many traditional DA methods lose their effectiveness due to this restricted access.

In this paper, we delve into the concept of source-free domain adaptation (SFDA) (Yu et al., 2023), where only a source domain pre-trained model and unlabeled target data are available for adaptation. Various types of SFDA methods have emerged, with a primary focus on data generation (Kurmi et al., 2021; Ding et al., 2022), self-training (Yang et al., 2022b; Litrico et al., 2023), and self-supervised learning (Zhang et al., 2022; Chen et al., 2022a). Data generation based methods encounter the challenge of accurately utilizing generated data to represent the unseen source domain. Meanwhile, self-training based methods are vulnerable to the issue of noisy pseudo labels. Additionally, self-supervised learning based methods come with added computational costs and necessitate the definition of pre-defined pretext tasks. Nonetheless, none of these methods explicitly address the domain divergence within the target domain during the adaptation of a pre-trained model to the target domain with theoretical analysis.

In light of this, we seek to gain a deeper understanding of the SFDA problem through theoretical analysis on the target domain empirical risk bound. Subsequently, we aim to extract some potentially valuable insights to inform the design of novel algorithms for this challenging scenario. We demonstrate that two key factors govern the target domain empirical risk in the SFDA problem: the empirical population risk associated with pseudo-labeled target data, and the domain divergence between pseudo-labeled target data and unlabeled target data. In this paper, we illustrate how to

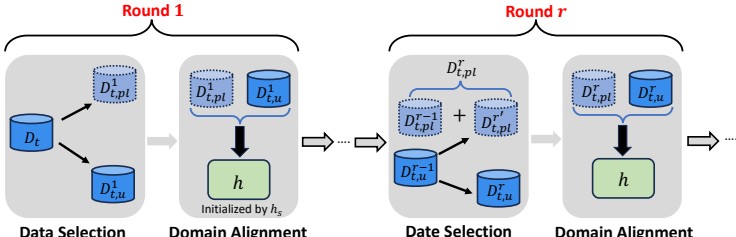

Figure 1: The proposed method overview. It is an iterative training process spanning across $R$ Rounds. Each round encompasses two fundamental stages: data selection and domain alignment. $D_t$ denotes the entirety of the unlabeled target dataset, with $D_{t,u}^r$ and $D_{t,pl}^r$ representing all unlabeled target data and pseudo-labeled target data during the $r$-th round, respectively. $D_{t,pl}^{r'}$ specifically refers to a selected subset of pseudo-labeled target data at the $r$-th round. Besides, $h$ is the target domain model that is initialized with source domain pre-trained model $h_s$ and adapted every round.

craft a novel algorithm guided by these two crucial insights to effectively tackle the SFDA problem. Specifically, we first propose a novel target data picking strategy to pseudo-label the samples with low label noise. Secondly, we present a semantic domain alignment strategy to gradually align unlabeled target data to the pseudo-labeled target data. Lastly, as shown in Figure 1, we introduce an iterative optimization scheme to combine the above two steps for multiple rounds.

Our method can be summarized for the following contributions:

- We propose a theoretical analysis on target domain empirical risk bound for SFDA problem, and derive the insights that we believe can inform the design of future SFDA algorithms.

- We introduce "Pick and Adapt", an effective solution for SFDA based on the insights. This iterative optimization regime jointly optimizes two key components: Top-k importance sampling and nearest neighbor voting-based semantic domain alignment.

- Extensive experiments across three widely applied DA benchmarks demonstrate the advantage of the proposed method for SFDA.

## 2 RELATED WORK

**Unsupervised Domain Adaptation (UDA).** Seminal works Ben-David et al. (2010) offer theoretical guarantees for quantifying the discrepancy between the source and target domains. The following works aim to either learn domain-invariant features or leverage adversarial learning to align and reduce the domain discrepancy. For example, Deep Adaptation Network (DAN) (Long et al., 2015) proposes to minimize the multi-kernel maximum mean discrepancy (MK-MMD) to reduce domain drift. The Joint Adaptation Network (Long et al., 2017) provides joint maximum mean discrepancy (JMMD) to align distribution across domains. Zhang et al. (2019) introduces Margin Disparity Discrepancy (MDD) to measure the divergence across domains. Adversarial Discriminative Domain Adaptation (ADDA) (Tzeng et al., 2017) combines discriminative modeling, untied weight sharing, and a GAN loss to learn generalizable features across domains. Conditional Domain Adversarial Networks (CDAN) (Long et al., 2018) aligns features across domains by exploiting discriminative information conveyed in the classifier predictions. Feature Gradient Distribution Alignment (FGDA) (Gao et al., 2021) employs a discriminator to differentiate the gradient distribution of features. Worth noting that all the methods need both source and target domain data for adaptation.

**Source-Free Domain Adaptation (SFDA).** An early work Source HypOthesis Transfer (SHOT) (Liang et al., 2020) proposes information maximization and clustering based pseudo-labeling to encourage the model for confident predictions on the target domain. Semantic Consistency on the Nearest Neighborhood (SCNNH) (Tang et al., 2021) shares a similar concept with SHOT, utilizing estimated cluster centroids to derive semantic information. Following efforts lie in two streams. One focuses on generating data to emulate the source domain, such as the Source Data free Domain Adaptation (SDDA) (Kurmi et al., 2021). The Distribution Estimation (DE) (Ding et al., 2022) focuses on estimating the class-conditioned feature distribution of the source domain by leveraging the target data and its corresponding anchor points. The other direction is self-training with the

label information provided by the pre-trained model, such as Neighborhood Reciprocity Clustering (NRC) (Yang et al., 2021) and Attracting and Dispersing (AaD) (Yang et al., 2022b). Litrico et al. (2023) improves the quality of pseudo labels by measuring their uncertainty. Divide and Contrast (DaC) Zhang et al. (2022) splits target data into source-like and target-specific groups. Different from all the above methods, our method roots from a rigorous theoretical analysis, where the population risk and unlabeled v.s. pseudo-labeled data domain drift are identified. For each of the problems, we specifically design the top-k importance sampling and nearest neighbor voting domain alignment to effectively minimize the empirical risk.

**Data Sampling (DS).** Data sampling aims to select partial representative or reliable data from whole dataset to improve the performance of learned model. For example, Posterior Sampling-based Outlier Mining (POEM)(Ming et al., 2022) picks the most informative outlier data from an extensive pool of auxiliary data points by selecting samples lying on the out-of-distribution decision boundary. Adaptive Outlier Optimization (AUTO)(Yang et al., 2023) introduces an in-out-aware filter designed to select and assign pseudo labels to data. The Importance Sampling method for Domain Adaptation (ISDA)(Xu et al., 2019) proposes a loss function incorporating feature-norm and prediction entropy to select data with significant information for effective domain. Cross-Domain Mixed Sampling (DACS)(Tranheden et al., 2021) involves selecting half of the classes in a source domain image, and then cutting out the corresponding pixels to paste them onto an image from the target domain. Recently, several representative data sampling strategies have been proposed for SFDA problem, for example, SHOT++ (Liang et al., 2021) utilizes the entropy values of predictions to partition the unlabeled target dataset into two subsets, treating them as a labeled subset and an unlabeled subset. ProxyMix (Ding et al., 2023) utilizes the weight of source pretrained classifier as the source class centers, and selects $N$ nearest data in target domain for each source class center as the labeled target data. Black-Box Model Adaptation by Domain Division (BETA) (Yang et al., 2022a) categorizes the unlabeled target dataset into two subsets, i.e., an easy-to-adapt subset and a hard-to-adapt subset by calculating probabilities for each target data. Unlike existing methods, our data sampling involves a gradual selection process throughout the training. In each round, we exclusively opt for target data with high-quality pseudo-labels.

## 3 METHODOLOGY

The problem notations are introduced as: $D_s = \{(x_s^i, y_s^i)\}_{i=1}^{n_s}$ denotes the labeled source domain dataset from source domain $\mathcal{D}_s$ with label space $\mathcal{C} = \{1, 2, ..., C\}$, and $D_t = \{x_t^i\}_{i=1}^{n_t}$ indicates the unlabeled target domain dataset from target domain $\mathcal{D}_t$. In this work, we demonstrate with the image classification task for the Source-free Domain Adaptation (SFDA) problem. Specifically, we target at adapting the source domain dataset $D_s$ pre-trained model $h_s(x) = (g_s \circ f_s)(x)$ to the unlabeled target domain dataset $D_t$, excluding the source data all through the process. The model includes a feature extractor $g_s : x \to z \in \mathbb{R}^D$, mapping images to a $D$-dimensional feature embedding, and a classifier $f_s : z \to p \in \mathbb{R}^C$, mapping the embedding to a $C$-dimensional probability vector.

### 3.1 THEORETICAL ANALYSIS ON SFDA

Based on the above setting, we first present a theoretical analysis on the target domain empirical risk bound for Source-free Domain Adaptation (SFDA). The derived bound further guides to a novel perspective for addressing this problem. Starting from a general empirical risk objective, we define it as the error between the target domain adapted model $h$ and the ground truth model $h_t$ on the target domain data $x_t \in D_t$:

$$\epsilon_t(h) = \mathbb{E}_{x_t \in D_t}\big[|h(x_t) - h_t(x_t)|\big], \tag{1}$$

Within the unlabeled target domain data, we assume that it is able to assign the ground truth label $y_t^i$ to a portion of the unlabeled target samples $x_t^i$. In this way, the target domain data can be divided into two subsets: the labeled subset denoted by $D_{t,l}$ and the unlabeled subset $D_{t,u}$. We further assume that $D_{t,l}$ and $D_{t,u}$ are i.i.d sampled from the target domain $\mathcal{D}_t$ with the size of $m$. As details shown in Appendix A.1, we present the following empirical risk bound on the target domain.

**Theorem 3.1.** *(proved in Appendix A.1) Consider a loss function $\epsilon(\cdot, \cdot)$ applied to a hypothesis $h$ and a dataset $D_t$ for empirical risk minimization. If $h$ is determined by the parameter $\theta$ and is trained on $D_t$, and it belongs to a hypothesis space $\mathcal{H}$ with a VC-dimension of $d$, with a probability of at least*

$1 - p$ *over the sample selection process, the following inequality holds:*

$$\epsilon_t(h) \leq 2\epsilon_{t,l}(h) + \frac{1}{2}\hat{d}_{\mathcal{H}\Delta\mathcal{H}}(D_{t,l}, D_{t,u}) + 2\sqrt{\frac{d\log(2m) + \log(\frac{2}{\delta})}{m}} + \lambda \quad (2)$$

*where $\hat{d}_{\mathcal{H}\Delta\mathcal{H}}(D_l, D_u)$ denotes the empirical distribution divergence, $m$ indicates the size of $D_{t,l}$, $\lambda$ indicates the classification error on $D_l$.*

From Theorem 3.1, we observe the main components to construct the risk bound, namely, the population risk $\epsilon_{t,l}(h)$ observed on $D_{t,l}$, the empirical domain drift between $D_{t,l}$ and $D_{t,u}$, and the $D_{t,l}$ size related variation. One challenge to computing the main components is: for population risk $\epsilon_{t,l}(h)$, we do not have ground truth supervision from the unlabeled target domain data. Following the self-supervised learning (SSL), we leverage the $h_s$, the source domain pre-trained model, as an alternative means to derive the pseudo label $y^i_{t,pl}$ on the target domain data $x^i_t$. Consequently, we assemble a pseudo-labeled target domain dataset, denoted as $D_{t,pl} = \{x^i_{t,pl}, y^i_{t,pl}\}^{n_{t,pl}}_{i=1}$. The empirical risk bound can be updated as the following:

$$\epsilon_t(h) \leq \underbrace{2\epsilon_{t,pl}(h)}_{\text{population risk}} + \underbrace{\frac{1}{2}\hat{d}_{\mathcal{H}\Delta\mathcal{H}}(D_{t,pl}, D_{t,u})}_{\text{domain divergence}} + \underbrace{2\sqrt{\frac{d\log(2m) + \log(\frac{2}{\delta})}{m}}}_{D_{t,pl}\text{size related variation}} + \lambda' \quad (3)$$

where $\lambda' = \lambda + \gamma$ and $\gamma$ is a constant error introduced by the quality of the pseudo-labels within $D_{t,pl}$. Equation 3 provides us a clear guidance that the target domain empirical risk for SFDA problem can be optimized via two components: (1) the empirical risk associated with $D_{t,pl}$, and (2) the empirical domain drift between $D_{t,pl}$ and $D_{t,u}$.

Based on Equation 3, regarding each component, we specifically design our method in the following sections. Section 3.2 tackles the first component, the population risk, by ensuring more accurate pseudo-labeling of the unlabeled target data. For the second component, Section 3.3 further demonstrates the proposed domain drift mitigation strategies. Section 3.4 summarizes the optimization objectives and the overall training regime.

## 3.2 TOP-K IMPORTANCE SAMPLING FOR PSEUDO LABELING

A canonical pseudo-labeling mechanism is to directly leverage the classifier $f$ from model $h$ to assign the class label, and conduct top-k sample selection as indicated below.

$$D^{\mathbf{C}}_{t,pl} = \bigcup_{c=1}^{C} \left\{ \text{Top}_K(D^c_{t,u}) \right\}, \; \text{Top}_K(D^c_{t,u}) = \{x^i_t | i \in \text{argsort}(h(x^i_t)), x^i_t \in D^c_{t,u}, i \leq K\}, \quad (4)$$

where $D^c_{t,u}$ denotes the unlabeled target data with pseudo label $y^i_{t,pl} = c$, and $K$ is the number of selected data. We denote this as the classifier-based sampling strategy (**C-sampling**). The vanilla C-sampling exhibits two main problems: **(1)** there is labeling bias inherited from the source domain pre-trained model $h_s$. **(2)** the selected samples with high confidence $h(D^c_{t,u})$ are mostly within source domain distribution, since deep learning trained classifiers are highly data-driven with strong distribution memorization. This enlarges the domain drift between $D_{t,pl}$ and $D_{t,u}$.

To mitigate the above issues, we leverage the unsupervised information by clustering the centers populated from the unlabeled target domain samples, which faithfully reflects the target domain distribution and pops up the most confident samples for pseudo-labeling. We term it as target class center based sampling strategy (**T-sampling**). Specifically, we apply the softmax over the logits after classifier $f$ to gauge the likelihood of correct classification, and select the top $\sigma\%$ of target data per class denoted as $D^{c,\sigma}_t$, to form a reliable set. This set is then utilized to estimate the target class centers through a weighted aggregation process:

$$\mu_c = \frac{\sum_{x^i_t \in D^{c,\sigma}_t} p^{i,c}_t \cdot z^i_t}{\sum_{x^i_t \in D^{c,\sigma}_t} p^{i,c}_t}, \quad (5)$$

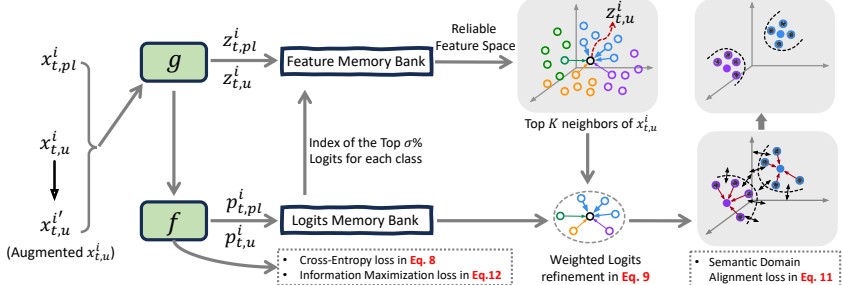

Figure 2: An illustration of the proposed semantic domain alignment via nearest neighbors voting.

where $p_t^{i,c}$ is the probability of the $x_t^i$ belongs to $c$-th class and $z_t^i$ is the feature of $x_t^i$. Following this, we calculate the cosine similarity between the estimated class centers $\mu_c$ and the unlabeled target data, and proceed to select data with high cosine similarity from each class:

$$D_{t,pl}^{\mathbf{T}} = \bigcup_{c=1}^{C} \left\{ \text{Top}_K^{cos}(D_{t,u}) \right\}, \ \text{Top}_K^{cos}(D_{t,u}) = \{x_t^i | i \in \text{argsort}(\cos(x_t^i, \mu_c)), x_t^i \in D_{t,u}, i \leq K\}. \tag{6}$$

Intuitively, the closer the unlabeled samples are to the estimated class centers, the higher the likelihood these samples belong to the specific classes. However, it is worth noting that the accuracy of the estimated class centers heavily relies on the pseudo label information generated by the model updated from the initial training rounds. This accuracy can be compromised due to inherent domain drift issues.

To further improve the pseudo-labeling accuracy, we introduce an intersection based sampling, denoted as **I-sampling**, to retrieve the intersection between C-sampling and T-sampling. With either of the two sampling sets empty, it degrades empty for the $c$-th category.

$$D_{t,pl}^{\mathbf{I}} = \bigcup_{c=1}^{C} \left\{ \text{Top}_K(D_{t,u}^c) \cap \text{Top}_K^{cos}(D_{t,u}) \right\}. \tag{7}$$

In Section 4.3, we empirically validate that **I-sampling** selects unlabeled target data with consistently higher pseudo-label purity compared to C-sampling and T-sampling.

The supervised cross-entropy loss is then applied to those pseudo labeled data to update the model $h$:

$$\mathcal{L}_{CE}(h; D_{t,pl}) = -\mathbb{E}_{(x_{t,pl}^i, y_{t,pl}^i) \in D_{t,pl}} \sum_{c=1}^{C} y_{t,pl}^{i,c} \log \sigma_c(h(x_{t,pl}^i)), \tag{8}$$

where $\sigma$ is the softmax operation, $\sigma_k$ indicates the $c$-th element in the softmax output of the model, and $y_{t,pl}^{i,c}$ denotes the $c$-th element in one-of-$C$ encoding of $y_{t,pl}^i$.

### 3.3 SEMANTIC DOMAIN ALIGNMENT VIA NEAREST NEIGHBORS VOTING

In this section, we primarily aim at aligning $D_{t,u}$ with the $D_{t,pl}$ obtained from Sec. 3.2 in order to reduce the domain discrepancy. Rather than directly applying the successful adversarial feature alignment (Goodfellow et al., 2014), we face a challenge that there is usually a significant size imbalance between $D_{t,pl}$ and $D_{t,u}$, which can result in additional adverse effects on domain alignment.

We alleviate the imbalance by spectating into the vicinity of each sample and re-balancing the volume by a uniform local density, e.g., the k nearest neighbors (KNN) (Taunk et al., 2019). As illustrated in Figure 2, consider an unlabeled target data $x_{t,u}^i$ drawn from $D_{t,u}$, along with a data augmentation operation denoted as $\text{Aug}(\cdot)$. We obtain the augmented data as $x_{t,u}^{i'} = \text{Aug}(x_{t,u}^i)$. We establish two memory banks: one for storing target features and the other for storing logits. The two memory banks are dynamically updated, with features and their respective logits being added in each mini-batch during the training process. For each unlabeled target data $x_{t,u}^i$, we use its feature $z_{t,u}^i$ to search the k nearest neighbors from the reliable set used in Equation 5. Consequently, the label information of the

$x_{t,u}^i$ is refined by aggregating the logits from the selected k nearest neighbors via a cosine similarity weighted mechanism:

$$\tilde{p}_{t,u}^i = \sum_{k=1}^K \alpha^k p_t^k \text{ and } \alpha^k = \frac{\exp\left(z_{t,u}^i \cdot z_t^k / \tau\right)}{\sum_{j=1}^K \exp\left(z_{t,u}^i \cdot z_t^j / \tau\right)}, \quad (9)$$

where $z_t^k$ denotes the feature of the nearest neighbors alone with its logits $p_t^k$. $\alpha^k$ measures the cosine similarity between the current data $x_{t,u}^i$ and the $k$-th nearest neighbor $x_t^k$. However, since the refinement is an iterative process, the refined label information by neighborhood label propagation can still be noisy. Inspired by (Fu et al., 2020), we propose a metric to indicate the certainty of the refined label information via:

$$\eta(\tilde{p}_{t,u}^i) = \frac{(1 - \mathtt{H}(\tilde{p}_{t,u}^i)) + \mathtt{Max}(\tilde{p}_{t,u}^i)}{2}, \quad (10)$$

where $\mathtt{H}(\cdot)$ denotes the entropy calculation and $\mathtt{Max}(\cdot)$ indicate the maximum probability value in $\tilde{p}_{t,u}^i$. We normalize the $\mathtt{H}(\tilde{p}_{t,u}^i)$ and $\mathtt{Max}(\tilde{p}_{t,u}^i)$ by minmax normalization to unify them within $[0, 1]$ before computing $\eta(\tilde{p}_{t,u}^i)$. This metric is designed to trade off the two ends. One is estimating certainty solely based on entropy may falter in distinguishing between confident and extremely sharp predictions. The other is relying solely on probability for certainty estimation may not effectively address the challenge of distinguishing between various class distributions.

With the refined label information $\tilde{p}_{t,u}^i$ and its certainty estimation $\eta(\tilde{p}_{t,u}^i)$, we align the domain drift between $D_{t,pl}$ and $D_{t,u}$ by a semantic alignment regularization, which pushes the data augmented $x_{t,u}^{i'} = \mathtt{Aug}(x_{t,u}^i)$ to the corresponding class centers learned by the model $h$:

$$\mathcal{L}_{SA}(h; D_{t,u}) = \mathbb{E}_{x_{t,u}^i \in D_{t,u}} \mathbb{I}\big(\eta(\tilde{p}_{t,u}^i) \geq \beta\big) \phi\Big(h\big(\mathtt{Aug}(x_{t,u}^i)\big), w^{\tilde{y}_{t,u}^i}\Big), \quad (11)$$

where $\beta$ is the threshold, $\phi(\cdot, \cdot)$ is any differentiable distance measure. We pick Wasserstein distance across all our experiments. $w$ is the row-wise neuron weights from the classifier $f$ viewed as the model learned class centers, and $\tilde{y}_{t,u}^i = \arg\max(\tilde{p}_{t,u}^i)$ is the label information for $x_{t,u}^i$.

Throughout the training process, a natural curriculum unfolds, driven by this regularization technique. Initially, the model generates lower-quality label information for $x_{t,u}^i$. Applying the $\beta$ parameter serves to filter out these unlabeled target data points with inferior label quality, thus preventing the risk of negative transfer. As training progresses, the unlabeled target data gradually aligns with the class centers. Consequently, higher-quality label information emerges, fostering improved domain alignment between the pseudo-labeled target data $D_{t,pl}$ and the unlabeled target data $D_{t,u}$.

### 3.4 Optimization Regime

As illustrated in Figure 1, our training pipeline unfolds dynamically across a span of $R$ rounds. Each round consists of two pivotal steps: pseudo labeled target data selection and semantic domain alignment via the nearest neighbor voting. To prevent the posterior collapse, we adhere to the well-known regularization technique, the Information Maximization (IM) loss:

$$\mathcal{L}_{IM}(h; D_{t,u}) = -\mathbb{E}_{x_{t,u}^i \in D_{t,u}} \sum_{c=1}^C \sigma_c\big(h(x_{t,u}^i)\big) \log\Big(\sigma_c\big(h(x_{t,u}^i)\big)\Big) + \sum_c^C P_c \log(P_c), \quad (12)$$

where $\sigma$ is the softmax operation, and $P_c = \mathbb{E}_{x_{t,u}^i \in D_{t,u}}[\sigma_c\big(h(x_{t,u}^i)\big]$. The overall objective for source-free domain adaptation is:

$$\mathcal{L}_{SFDA} = \mathcal{L}_{CE} + \mathcal{L}_{SA} + \rho \mathcal{L}_{IM}, \quad (13)$$

where $\rho$ is the loss balancing hyper-parameter.

## 4 Experiment

### 4.1 Experimental Settings

**Datasets**: We demonstrate our method under image classification task on three standard domain adaptation benchmarks as most of the literature does: Office-Home (Venkateswara et al., 2017),

Table 1: Classification Accuracy (%) on Office-Home (ResNet-50). The best results under SFDA setting are highlighted in bold. Note that "**SF**" means whether a method belongs to SFDA method.

| Method | SF | A→C | A→P | A→R | C→A | C→P | C→R | P→A | P→Cl | Pr→R | R→A | R→C | R→P | Avg. |
|--------|----|-----|-----|-----|-----|-----|-----|-----|------|------|-----|-----|-----|------|
| ERM | × | 34.9 | 50.0 | 58.0 | 37.4 | 41.9 | 46.2 | 38.5 | 31.2 | 60.4 | 53.9 | 41.2 | 59.9 | 46.1 |
| CDAN | × | 50.7 | 70.6 | 76.0 | 57.6 | 70.0 | 70.0 | 57.4 | 50.9 | 77.3 | 70.9 | 56.7 | 81.6 | 65.8 |
| CST | × | 59.0 | 79.6 | 83.4 | 68.4 | 77.1 | 76.7 | 68.9 | 56.4 | 83.0 | 75.3 | 62.2 | 85.1 | 73.0 |
| SHOT | ✓ | 57.1 | 78.1 | 81.5 | 68.0 | 78.2 | 78.1 | 67.4 | 54.9 | 82.2 | 73.3 | 58.8 | 84.3 | 71.8 |
| AaD | ✓ | 59.3 | 79.3 | 82.1 | 68.9 | 79.8 | 79.5 | 67.2 | 57.4 | 83.1 | 72.1 | 58.5 | 85.4 | 72.7 |
| DaC | ✓ | 59.1 | 79.5 | 81.2 | 69.3 | 78.9 | 79.2 | 67.4 | 56.4 | 82.4 | 74.0 | 61.4 | 84.4 | 72.8 |
| C-SFDA | ✓ | 60.3 | 80.2 | **82.9** | 69.3 | 80.1 | 78.8 | 67.3 | 58.1 | **83.4** | 73.6 | 61.3 | **86.3** | 73.5 |
| Ours | ✓ | **61.2** | **80.9** | 82.7 | 69.3 | **81.2** | **81.4** | **68.1** | 58.8 | 83.4 | **74.6** | 62.4 | 85.7 | **74.1** |

Table 2: Classification Accuracy (%) on DomainNet (ResNet-50). The best results under SFDA setting are highlighted in bold. Note that "**SF**" means whether the method belongs to SFDA method.

| Method | SF | Re→Cl | Re→Pa | Pa→Cl | Cl→Sk | Sk→Pa | Re→Sk | Pa→Re | Avg. |
|--------|----|-------|-------|-------|-------|-------|-------|-------|------|
| MCC | × | 44.8 | 65.7 | 41.9 | 34.9 | 47.3 | 35.3 | 72.4 | 48.9 |
| SHOT | ✓ | 67.7 | 68.4 | 66.9 | 60.1 | 66.1 | 59.9 | 80.8 | 67.1 |
| AC | ✓ | 70.2 | 69.8 | 68.6 | 58.0 | 65.9 | 61.5 | 80.5 | 67.8 |
| GPUE | ✓ | 74.2 | 70.4 | 68.8 | 64.0 | 67.5 | 65.7 | 76.5 | 69.6 |
| Ours | ✓ | **77.6** | **72.9** | **71.9** | **68.0** | **71.2** | **67.3** | **83.2** | **73.2** |

DomainNet (Peng et al., 2019), and VisDA-C 2017 (VisDA-C) (Peng et al., 2017). **(1) Office-Home** consists of four domains (Art, Clipart, Produce, and Real_world) with 65 classes. **(2) DomainNet** is one of the largest domain adaptation datasets which contains 6 domains and 345 classes. Following the protocol in (Litrico et al., 2023), we use a subset including 126 classes from 4 domains (Clipart, Painting, Real, Sketch), and evaluate on 7 designed domain tasks. **(3) VisDA-C** is a challenging large-scale synthesis-to-real object recognition dataset that contains 12 classes. The source domain includes $152k$ synthetic images and the target domain contains $55k$ real images. (The experimental results on VisDA-C can be found in the Appendix)

**Baselines:** We compare to two main streams of methods that are most relevant to ours: (1) *Unsupervised Domain Adaptation* methods, including CDAN (Long et al., 2018), MCC Jin et al. (2020), and CST (Liu et al., 2021). (2) *Source-Free Domain Adaptation* methods, including SHOT (Liang et al., 2020), AaD (Yang et al., 2022b), SDE (Ding et al., 2022), DaC (Zhang et al., 2022), GPUE (Litrico et al., 2023), and C-SFDA (Karim et al., 2023). Besides, we include Empirical Risk Minimization (ERM) (Koltchinskii, 2011) as a general baseline.

**Evaluation Metrics:** For Office-Home and DomainNet, we report the top-1 accuracy under each domain task together with their average. For VisDA-C, we report the per-class top-1 accuracy and their average. Each domain task is conducted independently 3 times and the average is reported.

**Implementation Details:** For a fair comparison, we adopt the pretraining methodology outlined in SHOT (Liang et al., 2020) to get the source domain pre-trained model. In particular, we separately employ ResNet50 for Office-Home and DomainNet, and ResNet101 for VisDA-C as the backbone architectures. Subsequently, we replace the original final FC layer with a new bottleneck layer, followed by Batch Normalization (BN). In the target domain adaptation phase, we initialize the target model using the parameters from the pretrained source domain model. We utilize SGD with momentum $0.9$ and weight decay $1e^{-3}$ and batch size of $64$ for all benchmarks. The initial learning rates are $0.001$ for Office-Home and DomainNet and $0.01$ for VisDA-C, respectively. We set $\rho = \exp{(ite/max\_ite)}^{-1}$ where *max_ite* is the maximum number of the training iterations. More implementation details can be found in Appendix.

Table 3: Classification Accuracy (%) on VisDA-C (ResNet-101). The best results under SFDA setting are highlighted in bold. Note that "**SF**" means whether the method belongs to SFDA method.

| Method | SF | plane | bcycl | bus | car | horse | knife | mcycl | person | plant | sktbrd | train | truck | Avg. |
|---|---|---|---|---|---|---|---|---|---|---|---|---|---|---|
| ERM | ✗ | 55.1 | 53.3 | 61.9 | 59.1 | 80.6 | 17.9 | 79.7 | 31.2 | 81.0 | 26.5 | 73.5 | 8.5 | 52.4 |
| CDAN | ✗ | 85.2 | 66.9 | 83.0 | 50.8 | 84.2 | 74.9 | 88.1 | 74.5 | 83.4 | 76.0 | 81.9 | 38.0 | 73.9 |
| MCC | ✗ | 88.1 | 80.3 | 80.5 | 71.5 | 90.1 | 93.2 | 85.0 | 71.6 | 89.4 | 73.8 | 85.0 | 36.9 | 78.8 |
| SHOT | ✓ | 94.3 | 88.5 | 80.1 | 57.3 | 93.1 | 94.9 | 80.7 | 80.3 | 91.5 | 89.1 | 86.3 | 58.2 | 82.9 |
| AaD | ✓ | 97.4 | 90.5 | 80.8 | 76.2 | 97.3 | 96.1 | 89.8 | 82.9 | 95.5 | 93.0 | 92.0 | 64.7 | 88.0 |
| DaC | ✓ | 96.6 | 86.8 | 86.4 | 78.4 | 96.4 | 96.2 | 93.6 | 83.8 | 96.8 | **95.1** | 89.6 | 50.0 | 87.3 |
| GPUE | ✓ | 97.3 | **96.2** | **90.5** | 91.8 | 90.0 | 94.2 | 87.4 | **87.7** | **97.0** | 84.3 | 93.0 | **81.0** | **90.0** |
| C-SFDA | ✓ | **97.6** | 88.8 | 86.1 | 72.2 | **97.2** | 94.4 | 92.1 | 84.7 | 93.0 | 90.7 | 93.1 | 63.5 | 87.8 |
| Ours | ✓ | 94.6 | 86.4 | 85.4 | **96.8** | 96.7 | 92.2 | **96.1** | 82.6 | 88.2 | 88.4 | 89.8 | 72.4 | 89.1 |

## 4.2 EXPERIMENTAL RESULTS

**Office-Home:** Table 1 illustrates the quantitative comparison on Office-Home dataset. The methods are organized into the top three rows, while ours is in the last row. The first row is the ERM result, which serves as the lower bound. The second row shows three representative unsupervised domain adaptation (UDA) methods, all of which necessitate the availability of the source domain dataset during adaptation to the target domain. The third row contains seven cutting-edge source-free domain adaptation (SFDA) methods, such as the SDE (Ding et al., 2022) and C-SFDA (Yang et al., 2022b). Under the UDA setting, our approach surpasses the strong baseline CST by $+1.1\%$ and outperforms all other UDA baselines by an even larger margin. This achievement is noteworthy considering that we do not utilize any source data during source-free adaptation. In the more challenging SFDA setting, we achieve a $2.2\%$ improvement over U-SFAN, a $1.2\%$ improvement over SDE, and a $0.6\%$ improvement over C-SFDA in terms of the 'Avg.' metric. Furthermore, our method outperforms all other SFDA baselines in 9 out of 12 domain tasks, firmly establishing its superiority.

**DomainNet:** The evaluation results on DomainNet are detailed in Table 2. We adhere to the experimental protocol outlined in (Litrico et al., 2023), which allows us to showcase the performance of our method across 7 different domain tasks. By comparing our method with both UDA and SFDA baselines, our method achieves state-of-the-art performance among all baselines and is higher than the second best GPUE by a margin of $3.6\%$ in terms of 'Avg.'. Furthermore, our method attains the highest accuracy across all 7 domain tasks. The achievement not only demonstrates the superiority of our proposed method but also underscores the effectiveness of our proposed approach in addressing source-free domain adaptation within the context of a large-scale benchmark.

**VisDA-C:** Table 3 presents a comparison between our method and state-of-the-art UDA and SFDA approaches on the VisDA-C dataset, considering the synthetic-to-real shift. Following the similar layout in Table 1, the top three rows show methods of ERM, UDA, and SFDA methods, with ERM serving as the lower bound. In this evaluation, we consistently observe significant improvement over most of the compared approaches. We achieve a performance increase of $1.9\%$ on average compared to CAN, without the need for access to source data during adaptation. Furthermore, our method achieves comparable performance to the best SFDA baseline, GPUE, while surpassing all other state-of-the-art SFDA methods by a considerable margin, such as a $1.1\%$ advantage over AaD, a $2.3\%$ improvement over AC, and a $1.3\%$ enhancement over C-SFDA, as measured by the 'Avg.'.

## 4.3 ANALYSIS AND DISCUSSIONS

**Ablation Study:** Table 4 shows the ablative results regarding the proposed modules. It is organized into three rows. In the first row, we individually explore each module within our method. Notably, the $\mathcal{L}_{SA}$ module exhibits the lowest accuracy at $28.2\%$ compared to the other two modules. This discrepancy arises from the fact that training the model exclusively with the selected $D_{t,pl}$ dataset or failing to prevent poster collapse would lead to degradation from the original source domain pre-trained model. This is because $D_{t,pl}$ is relatively small and contains incorrect label information. In the second row, we consistently observe improvements in accuracy when randomly combining two sub-modules. Notably, when we combine $\mathcal{L}_{SA}$ with either $\mathcal{L}_{CE}$ or $\mathcal{L}_{IM}$, we achieve a significant accuracy boost by more than $40.0\%$. Finally, in the last row, we showcase the performance enhancement achieved by our full approach, which further improves accuracy by more than $2.0\%$.

Table 4: Ablation study of sub-modules in our proposed method on Office-Home and DomainNet datasets with six domain tasks, i.e., A→C, A→P, A→R, Re→Cl, Re→Pa, and Re→Sk. $\mathcal{L}_{CE}$ is the cross-entropy loss for $D_{t,pl}$. $\mathcal{L}_{SA}$ is the semantic domain alignment loss for $D_{t,u}$. $\mathcal{L}_{IM}$ is the information maximization loss for $D_{t,u}$.

| $\mathcal{L}_{SA}$ | $\mathcal{L}_{CE}$ | $\mathcal{L}_{IM}$ | A→C | A→P | A→R | Re→Cl | Re→Pa | Re→Sk | Avg. |
|---|---|---|---|---|---|---|---|---|---|
| × | × | ✓ | 55.9 | 75.4 | 77.6 | 66.8 | 68.1 | 59.0 | 67.1 |
| × | ✓ | × | 56.9 | 73.7 | 78.5 | 66.1 | 62.5 | 57.2 | 65.8 |
| ✓ | × | × | 12.7 | 33.3 | 40.7 | 24.2 | 38.6 | 19.9 | 28.2 |
| × | ✓ | ✓ | 58.7 | 75.7 | 81.1 | 70.2 | 67.9 | 60.7 | 69.0 |
| ✓ | × | ✓ | 56.3 | 76.7 | 80.6 | 69.5 | 66.6 | 63.5 | 68.9 |
| ✓ | ✓ | × | 59.5 | 78.7 | 80.5 | 74.9 | 69.7 | 64.4 | 71.3 |
| ✓ | ✓ | ✓ | **61.2** | **80.9** | **82.7** | **77.6** | **72.9** | **67.3** | **73.8** |

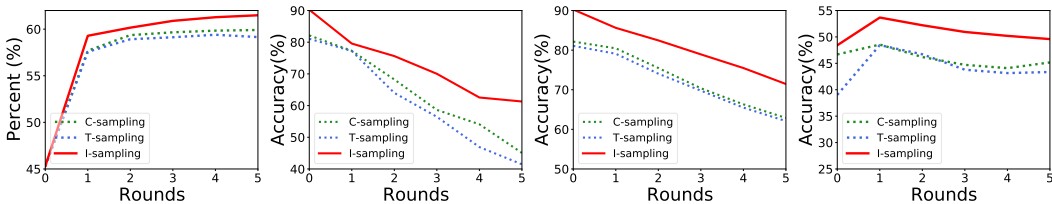

(a) Accuracy on $D_t$  (b) Accuracy on $D_{t,pl}^{r'}$  (c) Accuracy on $D_{t,pl}^{r}$  (d) Accuracy on $D_{t,u}^{r}$

Figure 3: As shown from left to right, four figures illustrate the accuracy of different target subsets during the training process with domain task A→C on Office-Home. (a) demonstrates the accuracy of all target data $D_t$. (b) shows the accuracy of the selected pseudo labeled target data $D_{t,pl}^{r'}$ at the $r$-th round. (c) describes the accuracy of the all pseudo labeled target data $D_{t,pl}^{r}$ at the $r$-th round. (d) plots the trend of the accuracy of the remaining unlabeled target data $D_{t,u}$ at the $r$-th round.

**Accuracy of the Target Data.** We explore the accuracy of the target data from different subsets, i.e., all target data $D_t$, selected pseudo labeled target $D_{t,pl}^{k'}$ at $r$ round, all pseudo labeled target data $D_{t,pl}^{k}$, and remaining unlabeled target data $D_{t,u}$. Figure 3 (a) presents the overall trend on $D_t$. To delve into the source of this advantage, we further examine the accuracy trend of two aspects: the selected pseudo labeled target data $D_{t,pl}^{r'}$ at each round, and the overall selected pseudo labeled target data $D_{t,pl}^{k}$. As depicted in Figures 3 (b) and (c), we observe that **I-sampling** strategy excels in selecting target data with more accurate labels, thereby enhancing model training. It's worth noting that as the data sampling process continues, the accuracy of selected target data with labels tends to decrease. This arises from the fact that the most reliable target data with pseudo-labels are chosen in the initial stages. Additionally, we see that the decline in accuracy for the selected pseudo labeled target data $D_{t,pl}^{r'}$ during the initial phase is less pronounced compared to the later stages. Conversely, the increase in accuracy for the entire target data $D_t$ during the early stages is more substantial than in the later stages. It is likely that accuracy on non-selected target data experiences significant improvement in the early training stages, leading to the selection of more reliable target data for labeling, which is verified in Figure 3 (d).

## 5 CONCLUSION

In this work, we conduct a theoretical analysis on the empirical risk bound for the Source-Free Domain Adaptation (SFDA) problem setting, where we identify two major factors to determine the bound, the pseudo-labeled population empirical risk and the pseudo-labeled and unlabeled subsets domain drift. We hope the theoretical analysis can inspire future new approaches from the identified factors. Subsequently, we propose a top-k importance sampling strategy to purify the pseudo-labeling process targeting for lower population risk, and a nearest neighbor voting-based semantic domain alignment approach to close the domain gap between the pseudo-labeled and remaining unlabeled target data. An iterative optimization is introduced to combine the two steps for multiple rounds. Across three major domain adaptation benchmarks, we achieve consistently better classification accuracy compared to the unsupervised domain adaptation and source-free domain adaptation methods.

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

# A   APPENDIX

The appendix contains in-depth information regarding the theoretical analysis, algorithm descriptions, implementation specifics, and supplementary experimental results.

## A.1   THEORETICAL DETAILS

A hypothesis is a function represented as $h : \mathcal{X} \to \{0, 1\}$. The probability according to the distribution $D$ that a hypothesis $h$ disagrees with a labeling function $f$ (which can also be a hypothesis) is defined as

$$\epsilon(h, f) = \mathbb{E}_{x \in D}\big[|h(x) - f(x)|\big]. \tag{14}$$

When we intend to denote the source error of a hypothesis associated with source domain $D_s$, we use the shorthand $\epsilon_s(h) = \epsilon_s(h, f_s)$. Further, we use the notation $\hat{\epsilon}_s(h)$ to represent the empirical source error. Similarly, for the target domain $D_t \triangleq \{x_t^i\}_{i=1}^{n_t}$, we employ the notations $\epsilon_t(h, f_t)$, $\epsilon_t(h)$, and $\hat{\epsilon}(h)$. Suppose that for some unlabeled target data $x_t^i \in D_t$, we can assign the ground truth label $y_t^i$ to them. Thus, the target domain can be divided into two subsets: the labeled subset denoted as $D_t^l \triangleq \{(x_{t,l}^i, y_{t,l}^i)\}_{i=1}^{n_t^l}$ and the unlabeled subset indicated as $D_t^u \triangleq \{x_{t,u}^i\}_{i=1}^{n_t^u}$.

**Definition A.1** (Based on Ben-David et al. (2010) ). *Given a domain $\mathcal{X}$ with $D$ and $D'$ probability distribution over $\mathcal{X}$, let $\mathcal{H}$ be a hypothesis class on $\mathcal{X}$ and denote by $I(h)$ the set for which $h \in \mathcal{H}$ is the characteristic function; that is, $x \in I(h)$,. The $\mathcal{H}$-divergence between $D$ and $D'$ is*

$$d_{\mathcal{H}}(D, D') = 2 \sup_{h \in \mathcal{H}} \big|Pr_D(I(h)) - Pr_{D'}[I(h)]\big| \tag{15}$$

**Lemma A.1** (Based on Ben-David et al. (2010) ). *Let $\mathcal{H}$ be a hypothesis space on $\mathcal{X}$ with VC dimension $d$, If $\mathcal{U}$ and $\mathcal{U}'$ are samples of size $m$ from $D$ and $D'$ respectively and $\hat{d}_{\mathcal{H}}(\mathcal{U}, \mathcal{U}')$ is the empirical $\mathcal{H}$-divergence between samples, then for any $\delta \in (0, 1)$, with probability at least $1 - \delta$,*

$$d_{\mathcal{H}}(D, D') \le \hat{d}_{\mathcal{H}}(\mathcal{U}, \mathcal{U}') + 4\sqrt{\frac{d \log{(2m)} + \log{(\frac{2}{\delta})}}{m}} \tag{16}$$

**Lemma A.2.** *For any hypothesis $h, h' \in \mathcal{H}$,*

$$\begin{aligned}
\big|\epsilon_s(h, h') - \epsilon_t(h, h')\big| &\le \sup_{h,h' \in \mathcal{H}} \big|\epsilon_s(h, h') - \epsilon_t(h, h')\big| \\
&= \sup_{h,h' \in \mathcal{H}} \big|Pr_{x \in D_s}[h(x) \neq h'(x)] - Pr_{x \in D_t}[h(x) \neq h'(x)]\big| \\
&= \frac{1}{2} d_{\mathcal{H}\Delta\mathcal{H}}(D_s, D_t)
\end{aligned}$$

**Theorem A.3.** *Given an unlabeled target domain $D_t$, we can assign the ground truth label $y_t^i$ to some unlabeled target data $x_t^i$. Thus, the target domain can be divided into two subsets: the pseudo-labeled subset denoted as $D_{t,l}$ and the unlabeled subset indicated as $D_{t,u}$. We assume that $\mathcal{U}_{t,l}$ and $\mathcal{U}_{t,u}$ are i.i.d. induced from the $D_{t,l}$ and $D_{t,u}$ with size of $m$, respectively. Let $\epsilon(\cdot, \cdot)$ be a loss function on a hypothesis and a dataset (for empirical error) or a distribution (for generalization error). If $h$ is governed by the parameter $\theta$ trained on $D_t$ and belongs to a hypothesis space $\mathcal{H}$ of VC-dimension $d$, then with probability at least $1 - p$ over the choice of samples, the inequality holds,*

$$\epsilon_t(h) \le 2\epsilon_{t,l}(h) + \frac{1}{2}\hat{d}_{\mathcal{H}\Delta\mathcal{H}}(D_{t,l}, D_{t,u}) + 2\sqrt{\frac{d \log{(2m)} + \log{(\frac{2}{\delta})}}{m}} + \lambda \tag{17}$$

where $d_{\mathcal{H}\delta\mathcal{H}}(D_l, D_u)$ denotes the distribution divergence, and $\lambda = \min\{\epsilon_{t,l}(h, f_t), \epsilon_{t,u}(h, f_t)\}$.

*Proof.* Recall that $\epsilon_t(h) = \epsilon_t(h, f_t)$, and $D_t = \{D_{t,l}, D_{t,u}\}$. Similarly, we have $\epsilon_{t,l}(h) = \epsilon_{t,l}(h, f_{t,l})$ and $\epsilon_{t,u}(h) = \epsilon_{t,u}(h, f_{t,u})$.

$$\epsilon_t(h) = \mathbb{E}_{x_t \in D_t}\left[\left|h(x_t) - f(x_t)\right|\right] = \mathbb{E}_{x_t \in \{D_{t,l} + D_{t,u}\}}\left[\left|h(x_t) - f(x_t)\right|\right]$$

$$\leq \mathbb{E}_{x_{t,l} \in D_{t,l}}\left[\left|h(x_{t,l}) - f(x_{t,l})\right|\right] + \mathbb{E}_{x_{t,u} \in D_{t,u}}\left[\left|h(x_{t,u}) - f(x_{t,u})\right|\right] = \epsilon_{t,l}(h) + \epsilon_{t,u}(h)$$

$$= \epsilon_{t,l}(h) + \epsilon_{t,u}(h) + \epsilon_{t,l}(h) - \epsilon_{t,l}(h) + \epsilon_{t,l}(h, f_{t,u}) - \epsilon_{t,l}(h, f_{t,u})$$

$$= 2\epsilon_{t,l}(h) + \left(\epsilon_{t,l}(h, f_{t,u}) - \epsilon_{t,l}(h)\right) + \left(\epsilon_{t,u}(h) - \epsilon_{t,l}(h, f_{t,u})\right)$$

$$\leq 2\epsilon_{t,l}(h) + \left|\epsilon_{t,u}(h, f_{t,u}) - \epsilon_{t,l}(h, f_{t,u})\right| + \left|\epsilon_{t,l}(h, f_{t,u}) - \epsilon_{t,l}(h, f_{t,l})\right|$$

$$\leq 2\epsilon_{t,l}(h) + \sup_{h, f_{t,u} \in \mathcal{H}} \left|\epsilon_{t,u}(h, f_{t,u}) - \epsilon_{t,l}(h, f_{t,u})\right| + \left|\epsilon_{t,l}(h, f_{t,u}) - \epsilon_{t,l}(h, f_{t,l})\right|$$

$$= 2\epsilon_{t,l}(h) + \frac{1}{2}d_{\mathcal{H}\Delta\mathcal{H}}(D_{t,l}, D_{t,u}) + \left|\epsilon_{t,l}(h, f_{t,u}) - \epsilon_{t,l}(h, f_{t,l})\right|$$

$$\leq 2\epsilon_{t,l}(h) + \frac{1}{2}d_{\mathcal{H}\Delta\mathcal{H}}(D_{t,l}, D_{t,u}) + \epsilon_{t,l}(h, f_{t,u}) + \epsilon_{t,l}(h, f_{t,l})$$

$$\leq 2\epsilon_{t,l}(h) + \frac{1}{2}\hat{d}_{\mathcal{H}\Delta\mathcal{H}}(\mathcal{U}_{t,l}, \mathcal{U}_{t,u}) + 2\sqrt{\frac{d\log\left(2m\right) + \log\left(\frac{2}{\delta}\right)}{m}} + \lambda$$

The last step is an application of Lemma A.1 and A.2. $\lambda$ comes from the classification error on $\mathcal{D}_{t,l}$ with classifiers $f_{t,u}$ and $f_{t,l}$. □

## A.2 ALGORITHM

Algorithm 1 illustrates details of our proposed method.

---

**Algorithm 1** Our proposed Algorithm.

---

**Require:** Unlabeled target domain dataset $D_t$, initialized model $h$ by source domain pretrained model $h_s$.
**Ensure:** Learned model $h$.
1: **for** $r = 1$ to $R$ **do**                                     ▷ $R$ #Rounds
2:     **if** Mod($r$, $Q$)=0 **then**                              ▷ $Q$ rounds interval
3:         Apply Equation 7 for target data selection (Sec. 3.2)
4:     **end if**
5:     **for** $i$ to $I$ **do**                                     ▷ $I$ is the len of $D_t$
6:         Apply overall loss Equation 13 to train $h$ (Sec. 3.4)
7:     **end for**
8: **end for**

---

## A.3 DETAILS OF THE DATASETS AND IMPLEMENTATION

**Office-Home** is a challenging dataset, which includes 15,500 images from 65 categories in office and home circumstances, consisting of four particularly dissimilar domains: Artistic images (**A**), Clip Art (**C**), Product images (**P**), and Real-World images (**R**). We establish a total of 12 transfer tasks by incorporating all available domains. we configure the top-K sample parameter in Equation 7 to be 5, set the $\sigma$ in Equation 5 at $50\%$, setup top-k neighbors selected for target data refinement in Equation 9 to 5, and adjust the $\beta$ to 0.9. The training process contains 5 rounds, with each round consisting of 15 epochs.

**DomainNet** is a substantial domain adaptation dataset, notable for its extensive scale encompassing 6 domains and 345 classes. However, due to the presence of noisy labels in some domains and classes, we follow a specific protocol mentioned in Litrico et al. (2023). In line with this protocol, 4 domains (Real, Clipart, Painting, Sketch) and 125 classes are selected. We focus on the adaptation scenarios where the target domain is not real images, and construct 7 scenarios from the 4 domains. we configure the top-K sample parameter in Equation 7 to be 15, set the $\sigma$ in Equation 5 at $50\%$,

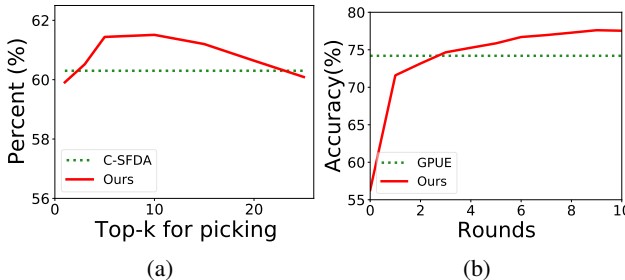

Figure 4: (a) Visualizing the influence of Top-k in Equation 7 on the picking pseudo labeled target data with domain task A→C on Office-Home. (b) Visualizing the training behavior of our method on adaptation task Re→Cl on DomainNet.

setup top-k neighbors selected for target data refinement in Equation 9 to 5, and adjust the $\beta$ to 0.9. The training process contains 10 rounds, with each round consisting of 10 epochs.

**VisDA-C** is a challenging large-scale synthesis-to-real object recognition dataset that contains 12 classes. The source domain includes $152k$ synthetic images and the target domain contains $55k$ real images. we configure the top-K sample parameter in Equation 7 to be 300, set the $\sigma$ in Equation 5 at $50\%$, setup top-k neighbors selected for target data refinement in Equation 9 to 200, and adjust the $\beta$ to 0.9. The training process contains 10 rounds, with each round consisting of 4 epochs.

## A.4 SENSITIVITY TO TOP-K IN EQUATION 7

To verify the impact of the Top-k in Equation 7, we conduct experiments on Office-Home with the adaptation task A→C. The value of the Top-k varies from 1 to 25. As shown in Figure 4 (a), We have observed that both a small and a large value for the Top-k lead to decreased performance in our study. In the case of a small Top-k value, the performance suffers due to the limited selection of pseudo-labeled data. This limitation negatively impacts the alignment between the pseudo-labeled and unlabeled target data, ultimately affecting the overall performance. Conversely, when employing a larger Top-k value, our method tends to select more data, including those with noisy label information. This abundance of noisy data adversely influences the performance, resulting in a decrease in overall effectiveness.

## A.5 ACCURACY VS. ROUND NUMBER CURVE FOR DOMAINNET

We delve deeper into understanding the training behavior of our approach on DomainNet. As depicted in Figure 4 (b), the accuracy trend of our method shows a gradual improvement. Notably, after four rounds, our approach demonstrates a significant performance boost, surpassing the state-of-the-art GPUE method (Litrico et al., 2023).

## A.6 HYPER-PARAMETER ANALYSIS

We evaluate the sensitivity of hyper-parameters in our method. Namely, the length of the round, the ratio of selected labeled data, the ratio of the selected reliable data, and the number of top-k data for labeling. As illustrated in Figure 5 (a) and Figure 6 (a), our performance exhibits continuous improvement with increasing round length. A longer round allows for more comprehensive model training, resulting in enhanced model performance. In Figure 5 (b) and Figure 6 (b), we observe the performance of our method across various ratios of target data selected as labeled data through pseudo-label assignment. The performance consistently improves within the range of $[0.1, 0.5]$. However, beyond a ratio of 0.5, there is a slight performance degradation. This decline is attributed to the inclusion of more target data as labeled data, which results in lower-quality pseudo-labels and consequently, a deterioration in model performance. As shown in Figure 5 (c) and Figure 6 (c),, we observe that a small ratio of selected reliable data results in lower performance. This is because a lower ratio of selected reliable data leads to a biased estimation of target class centers through the assigned pseudo-labels. In Figure 5 (d) and Figure 6 (d), we notice that both a small size of neighbors

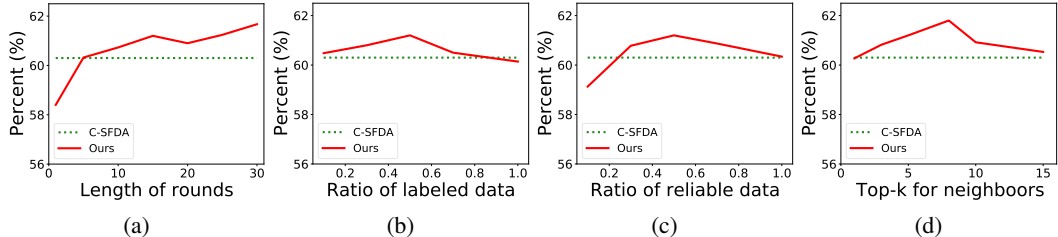

Figure 5: As shown from left to right, four figures provide insights into the effect of hyperparameters on our method when applied to the domain task A→C on Office-Home dataset. (a) illustrates the impact of the length of rounds. (b) delves into the impact of the ratio of target data selected as labeled data by assigning pseudo labels. (c) describes the effect of the $\sigma$ ratio of reliable data selected from the entire target data in Equation 5. (d) plots the influence of the top-k neighbors selected for target data refinement in Equation 9.

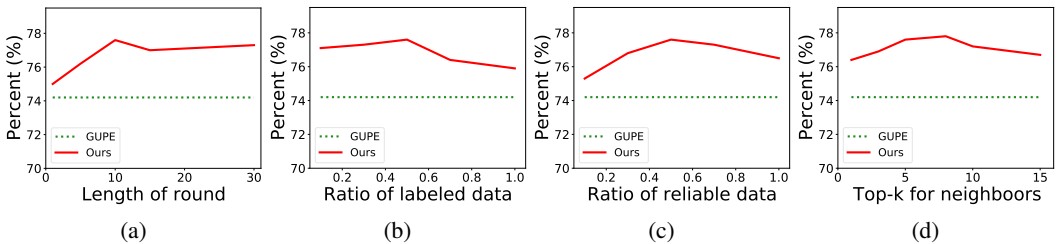

Figure 6: As shown from left to right, four figures provide insights into the effect of hyperparameters on our method when applied to the domain task Re→Cl on DomainNet dataset. (a) illustrates the impact of the length of rounds. (b) delves into the impact of the ratio of target data selected as labeled data by assigning pseudo labels. (c) describes the effect of the $\sigma$ ratio of reliable data selected from the entire target data in Equation 5. (d) plots the influence of the top-k neighbors selected for target data refinement in Equation 9.

and a large size of neighbors result in lower performance. A small neighbor size can introduce bias since there are fewer neighbors to contribute to label information. Conversely, a larger neighbor size can lead to a decrease in performance as it includes many dissimilar semantic neighbors, which in turn deteriorates the quality of the refined labels.

## A.7 PICKING AND ADAPTATION STRATEGIES STUDY

We integrated picking and adaptation strategies from two state-of-the-art methods, namely SHOT++ (Liang et al., 2021) and AaD Yang et al. (2022b). We denote our methods as "our-picking" + "our-adaptation". Specifically, we utilize the picking strategy from SHOT++, referred to as "SHOT++-picking," and the adaptation strategy from AaD, denoted as "AaD-adaptation." We evaluate the performance of "SHOT++-picking" + "our-adaptation" and "our-picking" + "AaD-adaptation" on adaptation tasks, namely A→C, A→P, and A→R, using the Office-Home dataset.

In Table 5 We observed that adopting the picking strategy from SHOT++, which uses entropy as the metric to select target data with entropy values larger than the average entropy values over the entire dataset as pseudo-labeled data, results in "SHOT++-picking" + "our-adaptation" significantly underperforming our results. This phenomenon can be attributed to two reasons. Firstly, "SHOT++-picking" heavily relies on the entropy value, which may struggle to distinguish between confident and extremely sharp predictions. Secondly, "SHOT++-picking" selects pseudo-labeled data only once, and the large number of selected data leads to a low quality of pseudo labels. Therefore, it demonstrates the advantage of our picking strategy compared to the picking strategy from SHOT++. Combining "our-picking" with "AaD-adaptation," we observe that it also underperforms compared to our approach. This is mainly due to "AaD-adaptation" blindly trusting the predicted semantic information of the neighbors, which can lead to negative clustering when these predictions are

Table 5: Picking and Adaption ablation study.

| Method | A→C | A→P | A→R |
|---|---|---|---|
| "SHOT++-Picking" + "our-adaptation" | 58.7 | 79.2 | 81.9 |
| "our-Picking" + "AaD-adaptation" | 60.1 | 79.7 | 82.3 |
| Ours | 61.2 | 80.9 | 82.7 |
| SHOT++ | 57.9 | 79.7 | 82.5 |
| AaD | 59.3 | 79.3 | 82.1 |

Table 6: Ablation study for data sampling.

| Sampling strategy | C-sampling | T-sampling | I-sampling(Ours) |
|---|---|---|---|
| A→C (OfficeHome) | 59.9 | 59.1 | 61.2 |
| Re→Cl (DomainNet) | 76.5 | 76.2 | 77.7 |

not very accurate. Additionally, the reliance on inaccurate local clusters can result in suboptimal discriminative representation learning. Therefore, it proves the advantage of our adaptation strategy over the adaptation strategy from AaD.

## A.8 ABLATION STUDY FOR C-SAMPLING AND T-SAMPLING

We adopt the adaptation tasks A→C and $Re \rightarrow Cl$ from OfficeHome and DomainNet, respectively, to investigate the performance of C-sampling and T-sampling. As depicted in Table 6, it is observed that the I-sampling strategy outperforms both C-sampling and T-sampling across various adaptation task.

## A.9 SENSITIVITY STUDY FOR $\rho$ IN EQ. 13

We have investigated the performance of our method across various values of $\rho$ ranging from 0.1 to 1.0 on the adaptation task $Re \rightarrow Cl$ from DomainNet. As indicated in Table 7, our method demonstrates lower sensitivity within the $\rho$ range of 0.3 to 1.0.

Table 7: Sensitivity study for $\rho$ in Eq. 13.

| $\rho$ | 0.1 | 0.3 | 0.5 | 0.8 | 1.0 |
|---|---|---|---|---|---|
| Re→Cl (DomainNet) | 75.9 | 77.0 | 77.2 | 77.4 | 77.5 |

