# OpenReview forum: "Pick and Adapt: An Iterative Approach for Source-Free Domain Adaptation"
_ICLR.cc/2024/Conference — Submitted to ICLR 2024_

### Official Review · Reviewer_hw5d · 2023-10-30

**Soundness:** 2 fair
**Presentation:** 2 fair
**Contribution:** 2 fair
**Rating:** 5
**Confidence:** 4

**Summary:**

This paper analyses the empirical risk bound for SFDA, and proposes a method that picks the reliable sets from target sets and then aligns the reliable pseudo-labeled sets with the remaining unlabeled sets.

**Strengths:**

- This method is easy to follow.
- The method achieves competitive results with the state-of-the-art methods.

**Weaknesses:**

The key techniques proposed in this manuscript are not novel. Some papers are highly correlated with the proposed method but not listed in the reference.
   - Picking a reliable set from target sets and then aligning the reliable pseudo-labeled sets with the remaining unlabeled sets.
   > [1] Source Data-absent Unsupervised Domain Adaptation through Hypothesis Transfer and Labeling Transfer. \
      [2] ProxyMix: Proxy-based mixup training with label refinery for source-free domain adaptation \
      [3] Divide to Adapt: Mitigating Confirmation Bias for Domain Adaptation of Black-Box Predictors \
      ...
   -  The neighbor aggregation strategy.
  > [1] Exploiting the Intrinsic Neighborhood Structure for Source-free Domain Adaptation. \
     [2] Do we really need to access the source data? source hypothesis transfer for unsupervised domain adaptation. \
     [3] Nearest neighborhood-based deep clustering for source data-absent unsupervised domain adaptation. \
     [4] Attracting and dispersing: A simple approach for source-free domain adaptation \
     ...
   - The theorem 3.1 and the proof are similar to [1].
   > [1] Learning Bounds for Domain Adaptation.
   - Besides, the sampling strategies and the curriculum learning regime are also not novel in 2023.

**Questions:**

- To validate the proposed techniques fairly, I think the author should compare the picking and adaption strategies separately with other well-known picking and adaption strategies. E.g., the minimum entropy, nearest distance to classifier weights, max probability, etc.
- I might improve the rating if the author could illustrate the novelty more clearly.

---

> ### Author Response · Authors · 2023-11-23
> **Response to Reviewer hw5d (Part 1)**
>
> ### **Q1: Difference between exisiting methods and ours.**
>
> **Response:** We sincerely thank the reviewer for providing insights into related works from different perspectives, including **data sampling** and **neighbor aggregation** strategies. We would like to clarify the differences between these methods and ours as follows. More details, please refer to the section: Related Work in revised version.
>
> **For data sampling,** there are 3 related papers suggested by the reviewer, i.e., Source Hypothesis Transfer (SHOT++) [1], Proxy-based Mixup (ProxyMix) [2], and Black-Box Model Adaptation by Domain Division (BETA) [3]. SHOT++ utilizes the entropy values of predictions to partition the unlabeled target dataset into two subsets, treating them as a labeled subset and an unlabeled subset. More precisely, it selects the top $t_k$ smallest entropy values for each class $k$. A significant challenge lies in the fact that the labeled subset constitutes half of the entire dataset. Additionally, it follows a two-step process, first splitting the data and then performing adaptation. ProxyMix relies on the weights of the source domain pretrained classifier as the source class prototypes. It then searches for the $N$ nearest samples in the target domain for each source class prototype (referred to as C sampling in our paper). Importantly, ProxyMix performs this selection only at the beginning and subsequently proceeds with the training. BETA categorizes the unlabeled target dataset into two subsets: an easy-to-adapt subset and a hard-to-adapt subset. This is achieved by calculating probabilities for each target data. The high probability target data are chosen as the easy-to-adapt subset, and their pseudo labels are set as the ground truth labels. The remaining data form the hard-to-adapt subset. Importantly, BETA conducts this selection only at the outset and then continues with the training process.
>
> Our selection strategy differs significantly from the aforementioned methods in several key aspects. **Firstly**, our approach involves a continual selection strategy. In each round, we exclusively choose target data with high-quality pseudo labels. In contrast, the methods mentioned above perform a single-time selection. **Secondly**, our method relies on semantic information from both the pretrained model (C sampling) and the target domain (T sampling) data to pick the reliable samples, where the reference methods are only based on semantic information from the source domain pre-trained model. **Lastly**, our selection process is integrated with model training, enhancing our method's ability to pick target data with high-quality pseudo labels. Conversely, the reference methods follow a two-step framework, conducting data selection in the initial step and subsequently proceeding with model training. In this initial step, they tend to select a larger number of target data, trusting their pseudo labels, which may result in lower-quality selected target data. Table 1 summarizes the component-wise differences between our method and others.
>
> Table1: A component-wise comparison of our proposed picking strategy to other related methods, such as SHOT++, ProxyMix, and BETA, reveals distinctive features and characteristics.
> |Task |Joint Picking and Training ?   | Picking based on Adapted Model ? | Picking based on Target Information ? |
> |------|:--------------------------------------:|:----------------------------------------------:|:---------------------------------------------------:|
> |SHOT++ [1]  | $\times$ | $\checkmark$ |  $\times$|
> |ProxyMix [2]  | $\times$ |$\checkmark$  |  $\times$ |
> |BETA [3]  | $\times$ | $\checkmark$ |  $\times$ |
> |Ours   |$\checkmark$  | $\checkmark$ |  $\checkmark$ |
>
> [1] Jian Liang et al. (2021). “Source data-absent unsupervised domain adaptation through hypothesis transfer and labeling transfer.” In IEEE TPAMI. \
> [2] Yuhe Ding et al. (2023). “Proxymix: Proxy-based mixup training with label refinery for source-free domain adaptation.” In NN. \
> [3] Jianfei Yang et al. (2022). “Divide to adapt: Mitigating confirmation bias for domain adaptation of black-box predictors.” In ICLR.

---

> ### Author Response · Authors · 2023-11-23
> **Response to Reviewer hw5d (Part 2)**
>
> **For neighbor aggregation,**  there are 3 related papers suggested by the reviewer, i.e., Neighborhood Reciprocity Clustering (NRC) [4],  Source Hypothesis Transfer(SHOT) [5], Semantic Consistency on the Nearest Neighborhood (SCNNH) [6], and Attracting and Dispersing (AaD) [7]. NRC aims to align each target data with its neighbors using carefully designed affinity values. However, the semantic information of the neighbors may not always guarantee consistency. This misalignment can have adverse effects, particularly in challenging adaptation tasks. SHOT employs a clustering-based method to enhance the quality of pseudo labels for the target dataset. It utilizes all unlabeled target data with pseudo labels to estimate class centroids, subsequently leveraging these centroids to refine the pseudo labels. However, SHOT encounters difficulties in distinguishing between different classes when class boundaries are exceptionally vague, and the distributions of different classes closely overlap. Additionally, constructing centroids for each class using all target data poses challenges, as not all predicted semantic information is accurate, leading to potential adverse effects on centroid construction due to misclassified data. SCNNH shares a similar concept with SHOT, utilizing estimated cluster centroids to derive semantic information. The distinction lies in the approach to obtaining pseudo labels. While SHOT directly assigns pseudo labels using the estimated centroids, SCNNH combines semantic information from a data point's neighbors to determine pseudo labels. AaD shares a similar concept with NRC by clustering target data to its neighbors. The key distinction is that AaD not only clusters target data with its neighbors but also endeavors to push the target data away from samples that potentially belong to different classes.
>
> Our neighbor aggregation strategy aims to enhance the quality of estimated semantic information for unlabeled target data, setting it apart from NRC and AaD, where neighbor aggregation primarily serves to achieve local clustering. In comparison to SHOT and SCNNH, which use estimated target class centroids for refinement, our approach diverges in three key aspects. **Firstly**, instead of refining pseudo labels through estimated centroids, we leverage the semantic information of neighbors. **Secondly**, to address the potential negative influence of neighbors with low-quality semantic information, we establish a reliable neighbor space by selecting the top $\rho\%$ of target data per class to form a trustworthy set. Each target data's neighbors are then chosen from this reliable set, mitigating the adverse effects that might arise from using all target data as the pool for neighbor selection, which is common in SHOT and SCNNH. **Finally**, to further mitigate the impact of noisy label information from the refined pseudo labels, we introduce a metric that gauges the certainty of the refined label information using entropy and probability.  Table 2 summarizes the component-wise differences between our method and others.
>
>
> Table2:A component-wise comparison of the neighbor strategy in our method and other related methods, i.e., NRC, SHOT, SCNNH, and AaD, reveals distinctive features and characteristics.
> |Task |Local Alignment ?   | Refine Pseudo Labels ? | Alleviate Negative Pseudo Labels ? |
> |------|:--------------------------------------:|:----------------------------------------------:|:---------------------------------------------------:|
> |NRC [4]   | $\checkmark$ | $\times$ |  $\times$|
> |SHOT [5]   | $\times$ |$\checkmark$  |  $\times$ |
> |SCNNH [6]   | $\times$ | $\checkmark$ |  $\times$ |
> |AaD [7]   | $\checkmark$ | $\times$ |  $\times$ |
> |Ours   |$\times$  | $\checkmark$ |  $\checkmark$ |
>
>
> [4] Shiqi Yang et al. (2021). “Exploiting the intrinsic neighborhood structure for source-free domain adaptation.” In NeurIPS. \
> [5] Jian Liang et al. (2020). “Do we really need to access the source data? source hypothesis transfer for unsupervised domain adaptation.” In ICML. \
> [6] Song Tang et al. (2021). “Nearest neighborhood-based deep clustering for source data-absent unsupervised domain adaptation.” In arXiv. \
> [7] Shiqi Yang et al. (2022). “Attracting and dispersing: A simple approach for source-free domain adaptation.” In NeurIPS.

---

> > ### Author Response · Authors · 2023-11-23
> > **Response to Reviewer hw5d (Part 3)**
> >
> > ### **Q2: About the novelty.**
> >
> > **Response:** Thanks for the valuable comments. In summary, our contributions are twofold. Firstly, we introduce a novel generalization bound (**Equation 3 in revised version**) for Source-Free Domain Adaptation (SFDA), incorporating considerations of population risk, domain divergence between labeled and unlabeled target data, and the size of the target labeled subset. Secondly, leveraging insights from this novel SFDA generalization bound, we address the SFDA problem through three key dimensions.
> >
> >
> > Firstly, we propose an innovative target data picking strategy aimed at selecting high-quality target data with pseudo-labels, effectively mitigating the impact of noise associated with pseudo-labels on population risk. This strategy involves splitting the target data into labeled and unlabeled subsets, enhancing the robustness of the SFDA framework.
> >
> > Secondly, we introduce a semantic domain alignment approach between labeled and unlabeled target data through nearest neighbors voting. Our novel nearest neighbors voting strategy refines the quality of predicted semantic information for unlabeled target data, incorporating a well-designed metric to indicate the certainty of the refined pseudo-labels. This alignment process focuses on associating unlabeled target data with high certainty to labeled target data, thereby reducing the domain divergence between the two.
> >
> > Lastly, we integrate the target data picking and model training into a cohesive framework. This framework systematically selects unlabeled target data with high-quality pseudo-labels, progressively incorporating them into the labeled target data subset. This iterative process effectively increases the size of the target labeled subset, consequently minimizing the SFDA generalization bound.

---

> ### Author Response · Authors · 2023-11-23
> **Response to Reviewer hw5d (Part 4)**
>
> ### **Q3: compare the picking and adaption strategies separately with other well-known picking and adaption strategies.**
>
> **Response:** Thank you for the valuable comment. We have integrated picking and adaptation strategies from two state-of-the-art methods, namely SHOT++ [1] and AaD [7]. We denote our methods as "our-picking" + "our-adaptation". Specifically, we utilize the picking strategy from SHOT++, referred to as "SHOT++-picking", and the adaptation strategy from AaD, denoted as "AaD-adaptation." We evaluate the performance of "SHOT++-picking" + "our-adaptation" and "our-picking" + "AaD-adaptation" on adaptation tasks, namely A$\rightarrow$C, A$\rightarrow$P, and A$\rightarrow$R, using the Office-Home dataset.  More details, please refer to the Appendix A.7 in our revised version.
>
>
> In Table 3 We observed that adopting the picking strategy from SHOT++, which uses entropy as the metric to select target data with entropy values larger than the average entropy values over the entire dataset as pseudo-labeled data, results in "SHOT++-picking" + "our-adaptation" significantly underperforming our results. This phenomenon can be attributed to two reasons. Firstly, "SHOT++-picking" heavily relies on the entropy value, which may struggle to distinguish between confident and extremely sharp predictions. Secondly, "SHOT++-picking" selects pseudo-labeled data only once, and the large number of selected data leads to a low quality of pseudo labels. Therefore, it demonstrates the advantage of our picking strategy compared to the picking strategy from SHOT++. Combining "our-picking" with "AaD-adaptation," we observe that it also underperforms compared to our approach. This is mainly due to "AaD-adaptation" blindly trusting the predicted semantic information of the neighbors, which can lead to negative clustering when these predictions are not very accurate. Additionally, the reliance on inaccurate local clusters can result in suboptimal discriminative representation learning. Therefore, it proves the advantage of our adaptation strategy over the adaptation strategy from AaD.
>
> |Method |A$\rightarrow$C |A$\rightarrow$P |A$\rightarrow$R|
> |------|:--------------------------------------:|:----------------------------------------------:|:---------------------------------------------------:|
> |"SHOT++-Picking'' + "our-adaptation'' |58.7 |79.2 |81.9 |
> |"our-Picking'' + "AaD-adaptation'' | 60.1 |79.7 |82.3|
> |Ours |61.2 |80.9 |82.7 |
> |SHOT++ |57.9 |79.7 |82.5 |
> |AaD |59.3 |79.3 |82.1 |

---

> ### Author Response · Authors · 2023-11-23
> **Response to Reviewer hw5d (Part 5)**
>
> ### **Q4: About learning bounds, data sampling, and curriculum learning.**
>
> **Response:**  Thanks for the valuable comment. **For learning bounds:** while we acknowledge the resemblance between our theorem and its proof to those presented in [8], it's crucial to note that [8] specifically addresses the generalization bound for traditional domain adaptation, which may not be directly applicable to the context of source-free domain adaptation (SFDA). Drawing inspiration from \[8], we extend our efforts to establish a novel generalization bound tailored for SFDA. Building upon this bound, we subsequently propose a solution to address the challenges posed by the SFDA problem. **For  data sampling and curriculum learning:**  while data sampling strategies and curriculum learning have been extensively studied, the joint integration of these approaches to effectively address the source-free domain adaptation (SFDA) problem remains an intriguing solution. This is particularly noteworthy, guided by our proposed SFDA generalization bound
>
>
>
>
> [8] John Blitzer et al. (2007). “Learning bounds for domain adaptation.” In NeurIPS.

---

> > ### Author Response · Authors · 2023-11-23
> >
> > Dear Reviewer hw5d,
> >
> > Thanks again for the valuable comments. We have tried our best to clarify the concerns on the paper and have incorporated your valuable comments into the revised version. Please kindly let us know if there is anything unclear. We truly appreciate your feedback that helps us further improve this work.
> >
> > We hope our responses and updates facilitate your re-evaluation. Your insights are invaluable to us, and we deeply appreciate your time and attention.
> >
> > Warmest regards,
> >
> > Authors

---

### Official Review · Reviewer_tSbQ · 2023-10-30

**Soundness:** 2 fair
**Presentation:** 3 good
**Contribution:** 2 fair
**Rating:** 3
**Confidence:** 4

**Summary:**

Authors analyze Source-free domain adaptation tasks through an empirical risk bound under the assumption that some labels can be obtained. To minimize the risk bound which consists of the population risk term and domain divergence term, authors propose a pseudo labeling strategy and a domain alignment strategy. Experimental results across three domain adaptation datasets are provided to prove the effectiveness of two strategies above.

**Strengths:**

1. The paper gives a risk bound analysis for source free domain adaptation task.

2. Authors propose two strategies to minimize the population risk term and domain divergence term in the risk bound.

3. The structure of the paper is well organized.

**Weaknesses:**

1. The assumption of treating pseudo-labels directly as ground truth labels does not fit SFDA tasks, and it is unreasonable to assume that both D_tl and D_tu are both i.i.d sampled from the target doamin, so the correctness of Eq.2 is doubtful.

2. Authors should provide the ablation study about C-sampling and T-sampling to prove the effectiveness of I-sampling.

3. Ablation study should use diverse tasks, three tasks used in Table 3 are from the same source model.

4. Authors should analyze the sensitivity of trade off ρ in Eq. 13.

5. The proposed method is too complicated and does not correspond to the theoretical analysis of the risk bound.

**Questions:**

1. In Eq.7, if using the intersection between C-sampling and T-sampling, why it degrades to single sampling when the other set is empty?

2. The results of VISDA should be placed in the experimental section.

3. Figure 3 (b)(c) is hard to understand: Why accuracy of pseudo labels decrease from 90% during adaptation?

Typo error: the caption of Table 3, A->R.

---

> ### Author Response · Authors · 2023-11-23
> **Response to Reviewer tSbQ (Part 1)**
>
> ### **Q1: About the assumption of treating pseudo-labels directly as ground truth labels and i.i.d sampling from target domain.**
>
> **Response:** We respectfully disagree "treating pseudo-labels directly as ground truth labels does not fit SFDA tasks". "Treating pseudo-labels directly as ground truth labels" has been widely applied in source-free domain adaptation, early work such as the Source Hypothesis Transfer (SHOT) [1] utilizes a clustering-based approach to improve the quality of pseudo-labels for the unlabeled target dataset. Subsequently, it places trust in these pseudo-labels and applies cross-entropy loss to optimize the model.
> AdaContrast [2] also leverages weakly augmented unlabeled target data to obtain pseudo-labels. These pseudo-labels are then trusted as ground truth labels and combined with strongly augmented data for cross-entropy loss during the model update process. SFDA-DE [3] similarly leverages a source domain pre-trained model to assign pseudo-labels to unlabeled target domain data, and trust their pseudo-labels to optimize model during the training process. GPUE [4] employs a clustering-based pseudo-label refinement process to obtain higher-quality pseudo-labels. It then places trust in these pseudo-labels and utilizes cross-entropy loss to update the model.
>
> Regarding the assumption that both $D_{tl}$ and $D_{tu}$ are i.i.d sampled from the target domain, our approach aligns with similar insight found in many data selection based source-free domain adaptation papers. For instance, SHOT++ [5] divides the target dataset into two subsets—namely, the low-entropy subset and high-entropy subset—based on the entropy values of the model's predictions. These subsets can be treated as two distinct distributions due to their differing distributions alone with the entropy values. Proxy-based Mixup (ProxyMix) [6] also adopts a strategy of splitting the unlabeled target dataset. It achieves this by selecting the $N$ nearest samples in the target dataset for each source class prototype (derived from the source pre-trained classifier weights) as the labeled subset, while the remaining samples constitute the unlabeled subset. These two subsets are considered from both the target domain and the source proxy domain. Therefore, it's reasonable to assume $D_{tl}$ and $D_{tu}$ are i.i.d sampled from the target domain.
>
> [1] Jian Liang et al. (2020). “Do we really need to access the source data? source hypothesis transfer for unsupervised domain adaptation.” In ICML. \
> [2] Dian Chen et al. (2022).  “Contrastive test-time adaptation.” In CVPR. \
> [3] Ning Ding et al. (2022). “Source-free domain adaptation via distribution estimation.” In CVPR. \
> [4] Mattia Litrico et al. (2023). "Guiding pseudo-labels with uncertainty estimation for test-time adaptation." In CVPR. \
> [5] Jian Liang et al. (2021). “Source data-absent unsupervised domain adaptation through hypothesis transfer and labeling transfer.” In IEEE TPAMI. \
> [6] Yuhe Ding et al. (2023). “Proxymix: Proxy-based mixup training with label refinery for source-free domain adaptation.” In NN.
>
>
> ### **Q2: Ablation study for C-sampling and T-sampling.**
>
> **Response:** Thanks for the valuable suggestion. We adopt the adaptation tasks A$\rightarrow$C and Re$\rightarrow$Cl from OfficeHome and DomainNet, respectively. As depicted in Table 1, it is observed that the I-sampling strategy outperforms both C-sampling and T-sampling across various adaptation tasks. More detials, please refer to Appendix A.8 in our revised version.
>
> Table 1: ablation study for C-samping and T-sampling.
> |Sampling strategy |C-sampling |T-sampling |I-sampling(Ours) |
> |------|:--------------------------------------:|:----------------------------------------------:|:---------------------------------------------------:|
> |A$\rightarrow$C (OfficeHome) |59.9 |59.1 |61.2 |
> |Re$\rightarrow$Cl (DomainNet) |76.5 |76.2 |77.7  |

---

> > ### Author Response · Authors · 2023-11-23
> > **Response to Reviewer tSbQ (Part 2)**
> >
> > ### **Q3: Diverse tasks for ablation study.**
> >
> > **Response:**  *Response:** Thanks for the reviewer's valuable suggestion. We further adopt the adaptation tasks, i.e., Re$\rightarrow$Cl, Re$\rightarrow$Pa, and Re$\rightarrow$Sk, from DomainNet to do the ablation study. The results can be found in Table 2. More detials, please refer to Section 4.3: Analysis and Discussions in our revised version.
> >
> > Table 2: Ablation study of sub-modules in our proposed method on Office-Home and DomainNet datasets with six domain tasks. The three adaptation tasks i.e.,  A$\rightarrow$C, A$\rightarrow$P, and A$\rightarrow$P come from Office-Home. The remaining three adaptation tasks Re$\rightarrow$Cl, Re$\rightarrow$Pa, and Re$\rightarrow$Sk are from DomainNNet.
> > |$\mathcal{L}_{SA}$ | $\mathcal{L}_{CE}$ | $\mathcal{L}_{IM}$ |  A->C  |  A->P  |  A->R  |   Re->Cl  | Re->Pa  | Re->Sk  |Avg. |
> > |---------------------------|:--------------------------:|:--------------------------:|:--------:|:--------:|:---------:|:-----------:|:------------:|:----------:|:------:|
> > |$\times$ |$\times$ |$\checkmark$ |55.9 |75.4 |77.6 |66.8 |68.1 |59.0 |67.1 |
> > |$\times$ |$\checkmark$ |$\times$ |56.9 |73.7 |78.5 |66.1 |62.5 |57.2 |65.8  |
> > |$\checkmark$ |$\times$ |$\times$ |12.7 | 33.3 |40.7 |24.2 |38.6 |19.9 |28.2  |
> > |$\times$ |$\checkmark$ |$\checkmark$ |58.7 |75.7 |81.1 |70.2 |67.9 |60.7 |69.0 |
> > |$\checkmark$ |$\times$ |$\checkmark$ |56.3 |76.7  |80.6 |69.5 |66.6 |63.5 |68.9  |
> > |$\checkmark$ |$\checkmark$ |$\times$ |59.5 |78.7  |80.5 |74.9 |69.7 |64.4 |71.3 |
> > |$\checkmark$ |$\checkmark$ |$\checkmark$ |$\textbf{61.2}$ |$\textbf{80.9}$  |$\textbf{82.7}$ |$\textbf{77.6}$ |$\textbf{72.9}$ |$\textbf{67.3}$ |$\textbf{73.8}$|
> >
> > We also evaluate the sensitivity of hyper-parameters in our method on adaptation task, i.e., $Re\rightarrow Cl$, on DomainNet. Namely, the length of the round in Table 3, the ratio of selected labeled data in Table 4, the ratio of the selected reliable data in Table 5, and the number of top-k data for labeling in Table 6. We adopt the state-of-the-art source-free domain adaptation method, i.e., GPUE [1], as the baseline. More detials, please refer to Appendix A.6: Hyper-Parameter Analysis in our revised version.
> >
> >
> > Table 3: illustrates the impact of the length of round on adaptation task Re$\rightarrow$Cl on DomainNet dataset.
> > |Length of round (epoch) |1       |5     |10      |15     |20        |30        | GPUE [1] |
> > |--------------------------------|:-------:|:----:|:------:|:------:|:---------:|:-------:| :-------:|
> > |performance(%) |75.0 |76.2 |77.6 |77.0 |77.1 |77.3|74.2 |
> >
> >
> >
> >
> > Table 4: illustrates the impact of the ratio of labeled data on adaptation task Re$\rightarrow$Cl on DomainNet dataset.
> > |ratio of labeled data |0.1 |0.3 |0.5 |0.8 |1.0|GPUE [1] |
> > |--------------------------------|:-------:|:----:|:------:|:------:|:---------:| :-------:|
> > |performance(%) |77.1 |77.2 |77.6 |76.3 |75.9|74.2 |
> >
> >
> > Table 5: illustrates the impact of the ratio of reliable data on adaptation task Re$\rightarrow$Cl on DomainNet dataset.
> > |ratio of reliable data |0.1 |0.3 |0.5 |0.8 |1.0|GPUE [1] |
> > |--------------------------------|:-------:|:----:|:------:|:------:|:---------:| :-------:|
> > |performance(%) |75.2 |76.7 |77.6 |77.2 |76.4|74.2 |
> >
> > Table 6: illustrates the impact of the Top-K for neighbors on adaptation task Re$\rightarrow$Cl on DomainNet dataset.
> > |Top-K for neighbors |1 |3 |5 |8 |10 |15 |GPUE [1] |
> > |--------------------------------|:-------:|:----:|:------:|:------:|:---------:|:-------:| :-------:|
> > |performance(%) |76.4 |76.9 |77.6 |77.8 |77.1 |76.6|74.2 |
> >
> > [1] Mattia Litrico et al. (2023). "Guiding pseudo-labels with uncertainty estimation for test-time adaptation." In CVPR.

---

> ### Author Response · Authors · 2023-11-23
> **Response to Reviewer tSbQ (Part 3)**
>
> ### **Q4: Sensitivity of trad off $\rho$ in Eq.13..**
>
> **Response:**  Thank you for the valuable suggestion. We investigated the performance of our method across various values of $\rho$ ranging from $0.1$ to $1.0$ on the adaptation task Re$\rightarrow$Cl from DomainNet. As indicated in Table 7, our method demonstrates lower sensitivity within the $\rho$ range of 0.3 to 1.0.  More detials, please refer to Appendix A.9 in our revised version.
>
>
> Table 7:  sensitivity study for $\rho$ in Eq. 13 with adaptation task Re$\rightarrow$Cl on DomainNet dataset.
> |$\rho$ |0.1 |0.3 |0.5 |0.8 |1.0 |
> |--------------------------------|:-------:|:----:|:------:|:------:|:---------:|
> |$Re\rightarrow Cl$ (DomainNet) |75.9 |77.0 |77.2 |77.4 |77.5|
>
>
> ### **Q5: The connection between the proposed risk bound and the modules in our method.**
>
> **Response:** Our proposed method closely aligns with our theoretical analysis. In particular, the generalization bound (**Equation 3 in revised version**) primarily comprises three components: population risk, domain divergence between labeled and unlabeled target data, and the size of the target labeled subset. Drawing inspiration from these three components, our proposed method incorporates three key elements.
>
> **Firstly**, we propose an innovative target data picking strategy aimed at selecting high-quality target data with pseudo-labels, effectively mitigating the impact of noise associated with pseudo-labels on population risk. This strategy involves splitting the target data into labeled and unlabeled subsets, enhancing the robustness of the SFDA framework.
>
> **Secondly**, we introduce a semantic domain alignment approach between labeled and unlabeled target data through nearest neighbors voting. Our novel nearest neighbors voting strategy refines the quality of predicted semantic information for unlabeled target data, incorporating a well-designed metric to indicate the certainty of the refined pseudo-labels. This alignment process focuses on associating unlabeled target data with high certainty to labeled target data, thereby reducing the domain divergence between the two.
>
> **Lastly**, we integrate the target data picking and model training into a cohesive framework. This framework systematically selects unlabeled target data with high-quality pseudo-labels, progressively incorporating them into the labeled target data subset. This iterative process effectively increases the size of the target labeled subset, consequently minimizing the SFDA generalization bound.
>
> ### **Q6: If using the intersection between C-sampling and T-sampling, why it degrades to single sampling when the other set is empty?**
>
> **Response:**  Thanks for pointing out this point. What we try to express is that our I-sampling is the intersection between C-sampling and T-sampling, which is influenced by both C-sampling and T-sampling. If one of them becomes empty, I-sampling would be the empty for the corresponding category. We have revised this point in the updated version.
>
> ### **Q7: In Figure 3 (b)(c), why accuracy of pseudo labels decrease from 90% during adaptation?**
>
> **Response:** Thank you for your valuable feedback. Figures 3(b) and (c) illustrate the accuracy of pseudo labels during the $r$-th round. Specifically, (b) represents the accuracy of pseudo-labeled target data selected at the $r$-th round, while (c) displays the accuracy of all pseudo-labeled target data at the $r$-th round. During the $r$-th round, the selected pseudo-labeled target data generally exhibit higher-quality pseudo labels compared to the unselected target data obtained through C/T/I-sampling. Consequently, the accuracy of the pseudo-labeled data chosen during the $r$-th round tends to be higher than that of the overall pseudo-labeled data in the subsequent $r+1$-th round. This is because, in the $r+1$-th round, the selected pseudo-labeled target data is likely to have lower-quality pseudo labels compared with these selected in $r$-th round, as it originates from the remaining unlabeled target data. As a result, the accuracy of pseudo-labels would gradually decrease with the selection of more pseudo-labeled data.
>
> ### **Q8: Typo error and the place of the Table 4.**
>
> **Response:** Thank you for the valuable suggestion. We address the typo error and ensure that Table 4 is placed on the main page in the revised version.

---

> > ### Author Response · Authors · 2023-11-23
> >
> > Dear Reviewer tSbQ,
> >
> > Thanks again for the valuable comments. We have tried our best to clarify the concerns on the paper and added multiple experiments as requested. Please kindly let us know if there is anything unclear. We truly appreciate your feedback that helps us further improve this work.
> >
> > We hope our responses and updates facilitate your re-evaluation. Your insights are invaluable to us, and we deeply appreciate your time and attention.
> >
> > Warmest regards,
> >
> > Authors

---

### Official Review · Reviewer_PRg5 · 2023-10-31

**Soundness:** 3 good
**Presentation:** 3 good
**Contribution:** 2 fair
**Rating:** 6
**Confidence:** 4

**Summary:**

The paper focuses on the problem of source-free domain adaptation (SFDA) that is practically valuable and has attracted widespread attention. Following a theoretical analysis of SFDA that is provided in the paper, population risk and domain drift have been identified as major factors and a new method to address these aspects is proposed. More specifically the method performs iterative optimization and combines top-k importance sampling and nearest neighbour voting-based semantic segmentation domain alignment.

**Strengths:**

* Theoretical analysis of SFDA is provided and used to motivate the proposed approach.
* The proposed method generally performs well and improves state-of-the-art performance on Office-Home and DomainNet (not on VisDA-C though).
* Extensive comparison with other approaches on Office-Home and VisDA-C datasets (not on DomainNet though).
* There is ablation study to analyse the impact of the different components which is particularly useful as there are multiple components present.
* A very good analysis of hyperparameters is provided.

**Weaknesses:**

* The method is relatively more complex as there are several components involved that are needed to obtain strong performance.
* Ablation study would be better designed if various source datasets were considered (even if only three scenarios would be evaluated due to compute).
* DomainNet dataset has only a relatively small number of approaches evaluated.

**Questions:**

* How does the method compare to others in terms of adaptation time (especially with respect to ERM and the best performing competitors)?

---

> ### Author Response · Authors · 2023-11-23
> **Response to Reviewer PRg5 (Part 1)**
>
> ### **Q1: More dataset for ablation study.**
>
> **Response:** Thanks for the reviewer's valuable suggestion. We further adopt the adaptation tasks, i.e., Re$\rightarrow$Cl, Re$\rightarrow$Pa, and Re$\rightarrow$Sk, from DomainNet to do the ablation study. The results can be found in Table 1. More detials, please refer to Section 4.3: Analysis and Discussions in our revised version.
>
> Table 1: Ablation study of sub-modules in our proposed method on Office-Home and DomainNet datasets with six domain tasks. The three adaptation tasks i.e.,  A$\rightarrow$C, A$\rightarrow$P, and A$\rightarrow$P come from Office-Home. The remaining three adaptation tasks Re$\rightarrow$Cl, Re$\rightarrow$Pa, and Re$\rightarrow$Sk are from DomainNNet.
> |$\mathcal{L}_{SA}$ | $\mathcal{L}_{CE}$ | $\mathcal{L}_{IM}$ |  A->C  |  A->P  |  A->R  |   Re->Cl  | Re->Pa  | Re->Sk  |Avg. |
> |---------------------------|:--------------------------:|:--------------------------:|:--------:|:--------:|:---------:|:-----------:|:------------:|:----------:|:------:|
> |$\times$ |$\times$ |$\checkmark$ |55.9 |75.4 |77.6 |66.8 |68.1 |59.0 |67.1 |
> |$\times$ |$\checkmark$ |$\times$ |56.9 |73.7 |78.5 |66.1 |62.5 |57.2 |65.8  |
> |$\checkmark$ |$\times$ |$\times$ |12.7 | 33.3 |40.7 |24.2 |38.6 |19.9 |28.2  |
> |$\times$ |$\checkmark$ |$\checkmark$ |58.7 |75.7 |81.1 |70.2 |67.9 |60.7 |69.0 |
> |$\checkmark$ |$\times$ |$\checkmark$ |56.3 |76.7  |80.6 |69.5 |66.6 |63.5 |68.9  |
> |$\checkmark$ |$\checkmark$ |$\times$ |59.5 |78.7  |80.5 |74.9 |69.7 |64.4 |71.3 |
> |$\checkmark$ |$\checkmark$ |$\checkmark$ |$\textbf{61.2}$ |$\textbf{80.9}$  |$\textbf{82.7}$ |$\textbf{77.6}$ |$\textbf{72.9}$ |$\textbf{67.3}$ |$\textbf{73.8}$|
>
> We also evaluate the sensitivity of hyper-parameters in our method on adaptation task, i.e., $Re\rightarrow Cl$, on DomainNet. Namely, the length of the round in Table 2, the ratio of selected labeled data in Table 3, the ratio of the selected reliable data in Table 4, and the number of top-k data for labeling in Table 5. We adopt the state-of-the-art source-free domain adaptation method, i.e., GPUE [1], as the baseline. More detials, please refer to Appendix A.6: Hyper-Parameter Analysis in our revised version.
>
>
> Table 2: illustrates the impact of the length of round on adaptation task Re$\rightarrow$Cl on DomainNet dataset.
> |Length of round (epoch) |1       |5     |10      |15     |20        |30        | GPUE [1] |
> |--------------------------------|:-------:|:----:|:------:|:------:|:---------:|:-------:| :-------:|
> |performance(%) |75.0 |76.2 |77.6 |77.0 |77.1 |77.3|74.2 |
>
>
>
>
> Table 3: illustrates the impact of the ratio of labeled data on adaptation task Re$\rightarrow$Cl on DomainNet dataset.
> |ratio of labeled data |0.1 |0.3 |0.5 |0.8 |1.0|GPUE [1] |
> |--------------------------------|:-------:|:----:|:------:|:------:|:---------:| :-------:|
> |performance(%) |77.1 |77.2 |77.6 |76.3 |75.9|74.2 |
>
>
> Table 4: illustrates the impact of the ratio of reliable data on adaptation task Re$\rightarrow$Cl on DomainNet dataset.
> |ratio of reliable data |0.1 |0.3 |0.5 |0.8 |1.0|GPUE [1] |
> |--------------------------------|:-------:|:----:|:------:|:------:|:---------:| :-------:|
> |performance(%) |75.2 |76.7 |77.6 |77.2 |76.4|74.2 |
>
> Table 5: illustrates the impact of the Top-K for neighbors on adaptation task Re$\rightarrow$Cl on DomainNet dataset.
> |Top-K for neighbors |1 |3 |5 |8 |10 |15 |GPUE [1] |
> |--------------------------------|:-------:|:----:|:------:|:------:|:---------:|:-------:| :-------:|
> |performance(%) |76.4 |76.9 |77.6 |77.8 |77.1 |76.6|74.2 |
>
> [1] Mattia Litrico et al. (2023). "Guiding pseudo-labels with uncertainty estimation for test-time adaptation." In CVPR.

---

> ### Author Response · Authors · 2023-11-23
> **Response to Reviewer PRg5 (Part 2)**
>
> ### **Q2: About the baselines in DomainNet.**
>
> **Response:** Thank you for the valuable suggestions. In the case of the challenging DomainNet dataset, widely recognized in domain adaptation, we have opted to directly cite baseline results from the state-of-the-art method, GPUE [1]. Notably, DomainNet has only recently been employed for source-free domain adaptation tasks, resulting in a limited number of evaluated methods. However, our method demonstrates significantly improved performance compared to GPUE, underscoring the efficacy of our approach.
>
> ### **Q3: About the adaptation time.**
>
> **Response:** Thank you for the valuable suggestion. We evaluated the iteration time for the A$\rightarrow$C adaptation task in OfficeHome. Using a ResNet50 model with a batch size of $64$, our method incurs a cost of 0.52 second for each iteration. In comparison, ERM at the source pretrain stage costs 0.23 second per iteration, SHOT costs 0.31 second, and AaD costs 0.49 second. Overall, the adaptation time of our method is comparable to the best-performing competitors.

---

> > ### Author Response · Authors · 2023-11-23
> >
> > Dear Reviewer PRg5,
> >
> > Thanks again for the valuable comments. We have tried our best to clarify the concerns on the paper and added experiments as suggested. Please kindly let us know if there is anything unclear. We truly appreciate your feedback that helps us further improve this work.
> >
> > Best regards,
> >
> > Authors

---

### Official Review · Reviewer_UfbE · 2023-11-08

**Soundness:** 2 fair
**Presentation:** 3 good
**Contribution:** 2 fair
**Rating:** 6
**Confidence:** 4

**Summary:**

This paper proposes a simple but effective SFDA solution that focuses on the efficient use of unlabeled data, selecting the parts with rich information and distinct distribution for model optimization.
The theoretical analysis of this solution fully reveals an available SFDA theory, which utilizes top-k samples for reducing inner-class risk and presents a nearest neighbor voting for reducing distribution shift risk.

**Strengths:**

1. The proposed theoretical analysis exhibits a high degree of comprehensibility, rendering it accessible to researchers and practitioners alike.

2. The experimental results empirically demonstrate the efficacy of the proposed methodologies in enhancing SFDA performance across diverse benchmark datasets.

3. This paper is well-written, which is easy to understand and the intuition behind the proposed regularization-based is clear.

**Weaknesses:**

1. The proposed approach is based on the known techniques, using all data for training leads to information redundancy, which puts the model in a suboptimal state.
Training with "key samples" found by data sampling techniques can improve performance. This paradigm has been widely used [1][2][3][4].
Authors should consider introducing literature on similar paradigms in the related work.

[1] Ming Y, Fan Y, Li Y. Poem: Out-of-distribution detection with posterior sampling[C]//International Conference on Machine Learning. PMLR, 2022: 15650-15665.

[2] Yang P, Liang J, Cao J, et al. AUTO: Adaptive Outlier Optimization for Online Test-Time OOD Detection[J]. arXiv preprint arXiv:2303.12267, 2023.

[3] Xu X, He H, Zhang H, et al. Unsupervised domain adaptation via importance sampling[J]. IEEE Transactions on Circuits and Systems for Video Technology, 2019, 30(12): 4688-4699.

[4] Tranheden W, Olsson V, Pinto J, et al. Dacs: Domain adaptation via cross-domain mixed sampling[C]//Proceedings of the IEEE/CVF Winter Conference on Applications of Computer Vision. 2021: 1379-1389.

2. More ablation studies should be considered.
The difficulty of adaptation varies across different dataset combinations.
Existing ablation studies have only discussed Office-home A to C.

(1).  Do all adaptation tasks require 5 rounds to achieve a good result? What is the round number curve for other dataset combinations?

(2). The ratio of labeled data in different dataset combinations should be discussed, so as the " ratio of reliable data" and the " Top-k for neighbors"

(3). Actually, authors can count the number of samples that have been sampled, and even analyze whether there are some common samples (sampled multiple times), which will help to give more convincing analysis and conclusions.

**Questions:**

My main concern lies with the questions mentioned above concerning weaknesses. Could you kindly furnish me with elaborate responses to them? Having thorough answers to these queries might prompt me to reassess my evaluation.

---

> ### Author Response · Authors · 2023-11-23
> **Response to Reviewer UfbE  (Part 1)**
>
> ### **Q1: Data sampling related works.**
>
> **Response:** We appreciate the reviewer's insight regarding the effectiveness of training with "Training with "key samples" found by data sampling techniques can improve performance". The recommended works align closely with the thematic direction of our method. We incorporated these methods into our revised version. More details, please refer to the section: Related Work in our revised version..
>
> Posterior Sampling-based Outlier Mining (POEM) [1] and Adaptive Outlier Optimization (AUTO) [2] both address the Out-of-Distribution (OOD) detection problem. POEM strategically identifies the most informative outlier data from an extensive pool of auxiliary data points by selecting samples lying on the OOD decision boundary. In contrast, AUTO introduces an in-out-aware filter designed to assign pseudo labels to data. The Importance Sampling method for Domain Adaptation (ISDA) [3] and Domain Adaptation via Cross-Domain Mixed Sampling (DACS) [4] focus on unsupervised domain adaptation. ISDA proposes a novel loss function incorporating feature-norm and prediction entropy considerations to selectively identify data with significant information for effective domain alignment during the training process. On the other hand, DACS involves selecting half of the classes in a source domain image, and then cutting out the corresponding pixels to paste them onto an image from the target domain. In contrast to the aforementioned methods, our approach is distinct in its goal of selecting unlabeled target data with high-quality pseudo-labels.  Specifically, our data sampling involves a gradual selection process throughout the training. In each round, we exclusively opt for target data with high-quality pseudo-labels.
>
> [1] Yifei Ming et al. (2022). "Poem: Out-of-distribution detection with posterior sampling." In ICML. \
> [2] Puning Yang et al. (2023). "Auto: Adaptive outlier optimization for online test-time ood detection." In arXiv. \
> [3] Xuemiao Xu et al. (2019). "Unsupervised domain adaptation via importance sampling." In IEEE TCSVT. \
> [4] Wilhelm Tranheden et al. (2021). "Dacs: Domain adaptation via cross-domain mixed sampling." In WACV.

---

> ### Author Response · Authors · 2023-11-23
> **Response to Reviewer UfbE (Part 2)**
>
> ### **Q2: Ablation studies with different dataset.**
>
> **Response:** Thanks for the valuable suggestion. We further adopt the adaptation tasks, i.e., Re$\rightarrow$Cl, Re$\rightarrow$Pa, and Re$\rightarrow$Sk, from DomainNet to do the ablation study. The results can be found in Table 1. More detials, please refer to Section 4.3: Analysis and Discussions in our revised version.
>
> Table 1: Ablation study of sub-modules in our proposed method on Office-Home and DomainNet datasets with six domain tasks. The three adaptation tasks i.e.,  A$\rightarrow$C, A$\rightarrow$P, and A$\rightarrow$P come from Office-Home. The remaining three adaptation tasks Re$\rightarrow$Cl, Re$\rightarrow$Pa, and Re$\rightarrow$Sk are from DomainNNet.
> |$\mathcal{L}_{SA}$ | $\mathcal{L}_{CE}$ | $\mathcal{L}_{IM}$ |  A->C  |  A->P  |  A->R  |   Re->Cl  | Re->Pa  | Re->Sk  |Avg. |
> |---------------------------|:--------------------------:|:--------------------------:|:--------:|:--------:|:---------:|:-----------:|:------------:|:----------:|:------:|
> |$\times$ |$\times$ |$\checkmark$ |55.9 |75.4 |77.6 |66.8 |68.1 |59.0 |67.1 |
> |$\times$ |$\checkmark$ |$\times$ |56.9 |73.7 |78.5 |66.1 |62.5 |57.2 |65.8  |
> |$\checkmark$ |$\times$ |$\times$ |12.7 | 33.3 |40.7 |24.2 |38.6 |19.9 |28.2  |
> |$\times$ |$\checkmark$ |$\checkmark$ |58.7 |75.7 |81.1 |70.2 |67.9 |60.7 |69.0 |
> |$\checkmark$ |$\times$ |$\checkmark$ |56.3 |76.7  |80.6 |69.5 |66.6 |63.5 |68.9  |
> |$\checkmark$ |$\checkmark$ |$\times$ |59.5 |78.7  |80.5 |74.9 |69.7 |64.4 |71.3 |
> |$\checkmark$ |$\checkmark$ |$\checkmark$ |$\textbf{61.2}$ |$\textbf{80.9}$  |$\textbf{82.7}$ |$\textbf{77.6}$ |$\textbf{72.9}$ |$\textbf{67.3}$ |$\textbf{73.8}$|
>
> We also evaluate the sensitivity of hyper-parameters in our method on adaptation task, i.e., Re$\rightarrow$Cl, on DomainNet. Namely, the length of the round in Table 2, the ratio of selected labeled data in Table 3, the ratio of the selected reliable data in Table 4, and the number of top-k data for labeling in Table 5. We adopt the state-of-the-art source-free domain adaptation method, i.e., GPUE [1], as the baseline. More detials, please refer to Appendix A.6: Hyper-Parameter Analysis in our revised version.
>
>
> Table 2: illustrates the impact of the length of round on adaptation task Re$\rightarrow$Cl on DomainNet dataset.
> |Length of round (epoch) |1       |5     |10      |15     |20        |30        | GPUE [1] |
> |--------------------------------|:-------:|:----:|:------:|:------:|:---------:|:-------:| :-------:|
> |performance(%) |75.0 |76.2 |77.6 |77.0 |77.1 |77.3|74.2 |
>
>
>
>
> Table 3: illustrates the impact of the ratio of labeled data on adaptation task Re$\rightarrow$Cl on DomainNet dataset.
> |ratio of labeled data |0.1 |0.3 |0.5 |0.8 |1.0|GPUE [1] |
> |--------------------------------|:-------:|:----:|:------:|:------:|:---------:| :-------:|
> |performance(%) |77.1 |77.2 |77.6 |76.3 |75.9|74.2 |
>
>
> Table 4: illustrates the impact of the ratio of reliable data on adaptation task Re$\rightarrow$Cl on DomainNet dataset.
> |ratio of reliable data |0.1 |0.3 |0.5 |0.8 |1.0|GPUE [1] |
> |--------------------------------|:-------:|:----:|:------:|:------:|:---------:| :-------:|
> |performance(%) |75.2 |76.7 |77.6 |77.2 |76.4|74.2 |
>
> Table 5: illustrates the impact of the Top-K for neighbors on adaptation task Re$\rightarrow$Cl on DomainNet dataset.
> |Top-K for neighbors |1 |3 |5 |8 |10 |15 |GPUE [1] |
> |--------------------------------|:-------:|:----:|:------:|:------:|:---------:|:-------:| :-------:|
> |performance(%) |76.4 |76.9 |77.6 |77.8 |77.1 |76.6|74.2 |
>
> [1] Mattia Litrico et al. (2023). "Guiding pseudo-labels with uncertainty estimation for test-time adaptation." In CVPR.
>
>
> ### **Q3: Details of the rounds for all datasets.**
>
> **Response:** Thanks for the valuable comments. In all adaptation tasks within OfficeHone, the number of rounds has been standardized to 5. For DomainNet, a consistent setting of 10 rounds has been applied across all adaptation tasks. Similarly, for VisDA, we have chosen to implement 10 rounds as well. Namely, the length of the round, the ratio of selected labeled data, the ratio of the selected reliable data, and the number of top-k data for labeling. We also provide the accuracy vs. rounds curve for adaptation task Re$\rightarrow$Cl on DomainNet in Table 6. More detials, please refer to Appendix A.5: Accuracy vs. Round Number Curve for DomainNet in our revised version.
>
> Table 6: illustrates the accuracy vs. number of rounds on adaptation task Re$\rightarrow$Cl on DomainNet dataset.
> |Number of rounds          |0       |1        |2       |3       |4       |5       |6         |7        |8         |9        |        10|
> |--------------------------------|:-------:|:------:|:------:|:------:|:-----:|:-------:|:-------:|:-------:|:-------:|:-------:|:-------:|
> |performance(%)|56.3| 71.5| 73.1| 74.6| 75.2| 75.8| 76.7| 76.9| 77.2| 77.6| 77.5|

---

> ### Author Response · Authors · 2023-11-23
> **Response to Reviewer UfbE (Part 3)**
>
> ### **Q4:About the number of samples have been sampled.**
>
> **Response:**  We appreciate the reviewer's insightful recommendation for further analysis of our method. In our experiments, we have set the maximum ratio of labeled data to be $0.5$ across all datasets. Should the labeled data ratio reach $0.5$, subsequent picking operations will not be executed. Additionally, once a data in the target dataset is selected, it is excluded from the unlabeled subset, ensuring that no data is sampled multiple times.

---

> > ### Comment · Reviewer_UfbE · 2023-11-23
> > **Thanks for your reply**
> >
> > Thank you for your response. My question has been answered well, so I will raise the score from 5 to 6.

---

### Meta-Review · Area_Chair_HsFj · 2023-12-06

**Metareview:**

The assumption of treating pseudo-labels directly as ground truth labels does not fit SFDA tasks, and it is unreasonable to assume that both $D_{tl}$ and $D_{tu}$ are both i.i.d sampled from the target doamin, so the correctness of Eq.2 is doubtful.The proposed approach is based on the known techniques, using all data for training leads to information redundancy, which puts the model in a suboptimal state. Training with "key samples" found by data sampling techniques can improve performance. This paradigm has been widely used [1][2][3][4]. Authors should consider introducing literature on similar paradigms in the related work. Initially, some reviewers recommended reject, since the contribution of the paper is limited.

**Justification For Why Not Higher Score:**

This paper is of limited contribution and reviewers holds negetive evaluation of it. I read the paper too and find it not meets ICLR's requirement.

**Justification For Why Not Lower Score:**

N/A

---

### Decision · Program_Chairs · 2024-01-16

Reject